# Strong variation of spin-orbit torques with relative spin relaxation rates in ferrimagnets

Lijun Zhu [1,2] ✉ & Daniel C. Ralph [3,4]

Spin-orbit torques (SOTs) have been widely understood as an *interfacial* transfer of spin that is independent of the bulk properties of the magnetic layer. Here, we report that SOTs acting on ferrimagnetic $Fe_xTb_{1-x}$ layers decrease and vanish upon approaching the magnetic compensation point because the rate of spin transfer to the magnetization becomes much slower than the rate of spin relaxation into the crystal lattice due to spin-orbit scattering. These results indicate that the relative rates of competing spin relaxation processes within magnetic layers play a critical role in determining the strength of SOTs, which provides a unified understanding for the diverse and even seemingly puzzling SOT phenomena in ferromagnetic and compensated systems. Our work indicates that spin-orbit scattering within the magnet should be minimized for efficient SOT devices. We also find that the interfacial spin-mixing conductance of interfaces of ferrimagnetic alloys (such as $Fe_xTb_{1-x}$) is as large as that of 3*d* ferromagnets and insensitive to the degree of magnetic compensation.

Efficient manipulation of magnetic materials is essential for spintronic devices. While spin-orbit torques (SOTs)[1,2] are well established to be an effective tool to manipulate metallic 3*d* ferromagnets (FMs), whether they can effectively control antiferromagnetically-ordered systems has remained elusive despite the recent blooming of interest in ferrimagnets (FIMs) and antiferromagnets (AFs)[3–9]. Experimentally, for reasons unclear, the SOTs exerted on nearly compensated FIMs[5–7,10] are often measured to be considerably weaker than those on 3*d* FMs for a given spin-current generator (by up to >20 times, see below). More strikingly, it remains under debate whether perfectly compensated FIMs and collinear AFs ($M_s = 0$ emu/cm³) can be switched at all by SOTs[11–13].

Microscopically, SOTs have been widely assumed as an interfacial transfer of spin (i.e., spin dephasing length $\lambda_{dp} \approx 0$ nm for transverse spin current) that is independent of the bulk properties of the magnetic layer, such as in drift-diffusion analyses[14–16]. Under this assumption, spin current entering the magnet from an adjacent spin-generating layer is absorbed fully by the magnetization via dephasing to generate SOTs, and the damping-like SOT efficiency per current density ($\xi_{DL}^j$) will depend only on the spin Hall ratio ($\theta_{SH}$) of the spin-generating layer and the spin transparency ($T_{int}$) of the interface which determines what fraction of the spin current enters the magnet[17,18], i.e.,

$$\xi_{DL}^j = T_{int}\theta_{SH}. \tag{1}$$

This picture is a reasonable approximation for sufficiently thick metallic FMs that have a short $\lambda_{dp}$ (≤1 nm) due to strong exchange coupling[19–21] and a long spin diffusion length ($\lambda_s$) associated with spin relaxation due to spin-orbit scattering[22,23]. However, in antiferromagnetically-ordered systems $\lambda_{dp}$ can be quite long, as predicted more than a decade ago[24–27], which, as discussed below, questions the widely accepted models of "interfacial torques", particularly, in FIMs with strong spin-orbit scattering. So far, any roles of the bulk properties of the magnetic layer, e.g., the competing spin relaxation rates, in the determination of $\xi_{DL}^j$ have been overlooked in SOT analyses.

Here, we report measurements of SOTs acting on ferrimagnetic $Fe_xTb_{1-x}$ layers with strong spin-orbit coupling (SOC)[8] by tuning the $Fe_xTb_{1-x}$ composition and temperature (*T*). We find that, in contrast to

¹State Key Laboratory of Superlattices and Microstructures, Institute of Semiconductors, Chinese Academy of Sciences, Beijing 100083, China. ²College of Materials Science and Opto-Electronic Technology, University of Chinese Academy of Sciences, Beijing 100049, China. ³Cornell University, Ithaca, NY 14850, USA. ⁴Kavli Institute at Cornell, Ithaca, NY 14850, USA. ✉e-mail: ljzhu@semi.ac.cn

the prediction of Eq. (1), $\xi_{DL}^j$ varies strongly with the degree of magnetic compensation for a given $T_{int}$, due to changes in the fraction of spin current that relaxes directly to the lattice via SOC instead of being absorbed by the magnetization to apply SOTs. These results uncover the critical role of spin relaxation rates of the magnetic layer and provide a unified understanding for the diverse SOT phenomena in ferromagnetic and antiferromagnetically-ordered systems.

## Results and discussion
### Sample details
For this work, we sputter-deposited $Pt_{0.75}Ti_{0.25}$ (5.6 nm)/$Fe_xTb_{1-x}$ (8 nm) bilayers with different Fe volumetric concentrations ($x = 0.3-1$). The $Pt_{0.75}Ti_{0.25}$ layer, a dirty-limit Pt alloy with strong intrinsic spin Hall effect[17], sources spin current that exerts SOT on the FIM $Fe_xTb_{1-x}$. Each sample was deposited by co-sputtering on an oxidized Si substrate with a 1 nm Ta seed layer, and protected by a 2 nm MgO and a 1.5 nm Ta layer that was oxidized upon exposure to atmosphere. For electrical measurements, the samples were patterned into $5 \times 60\ \mu m^2$ Hall bars by photolithography and ion milling with a water-cooled stage. After processing, the magnetization hysteresis of the $Fe_xTb_{1-x}$ measured from the anomalous Hall voltage ($V_{AH}$) in patterned Hall bars shows fairly close coercivity (perpendicular depinning field) and squareness as the magnetization of unpatterned regions of the films measured by a superconducting quantum interference device (see Fig. 1a, b and Method). As shown in Fig. 1a–d, the $Fe_xTb_{1-x}$ has strong bulk perpendicular magnetic anisotropy (PMA) for $0.3 \le x \le 0.62$ and well-defined in-plane magnetic anisotropy for $0.75 \le x \le 1$. All the PMA samples have large anisotropy fields (14.4–72.2 kOe, as estimated from the fits in Supplementary Fig. S2a) and square hysteresis loops for both the out-of-plane magnetization and anomalous Hall voltage.

### Strong variation of spin-orbit torques with composition and temperature
We performed harmonic Hall voltage response (HHVR) measurements[28,29] by carefully separating out any potential thermoelectric effects (see Supplementary Note 2 for details). We choose the HHVR technique with excitation of a sinusoidal electric field $E$ (typically 30 kV/m) because it allows very accurate, consistent determination of SOTs for both IMA and PMA samples[30–32] without introducing significant thermal heating (Fig. 1a, b and Supplementary Fig. S4). We calculate the SOT efficiency using $\xi_{DL}^j = (2e/\hbar)M_s t_{FeTb} H_{DL}/j_c$[18], where $e$ is the elementary charge, $\hbar$ the reduced Plank's constant, $t_{FeTb}$ the $Fe_xTb_{1-x}$ thickness, $M_s$ the saturation magnetization of the $Fe_xTb_{1-x}$ (Supplementary Note 1), and $j_c = E/\rho_{xx}$ is the sinusoidal current density in the $Pt_{0.75}Ti_{0.25}$ with resistivity $\rho_{xx}$ ($j_c \approx 2.2 \times 10^6 A/cm^2$ for $E = 30$ kV/m). $H_{DL}$ is the current-driven damping-like SOT field. The "planar Hall correction" is negligible for these PMA $Fe_xTb_{1-x}$ samples ($V_{Ph}/V_{AH} < 0.1$, Supplementary Fig. S3).

In Fig. 2a, b, we show the measured values of $M_s$ and $\xi_{DL}^j$ at 300 K for the $Pt_{0.75}Ti_{0.25}$/$Fe_xTb_{1-x}$ bilayers with different $Fe_xTb_{1-x}$ compositions (we refer to this as the composition series). $M_s$ decreases monotonically by a factor of 33, from 1560 emu/cm³ for $x = 1$ (pure Fe, 3$d$ FM) to 47 emu/cm³ for $x = 0.5$ (nearly full compensation), and then increases slowly as $x$ further decreases. More details about the composition dependent magnetic properties are shown in Supplementary Note 1. As $x$ decreases in the Fe-dominated regime ($x \ge 0.5$), $\xi_{DL}^j$ decreases by a factor of 7 at 300 K, first slowly from $0.38 \pm 0.02$ for $x = 1$ to $0.27 \pm 0.01$ for $x = 0.61$ and then more rapidly to $0.054 \pm 0.002$ for $x = 0.5$. $\xi_{DL}^j$ increases slowly with decreasing $x$ in the Tb-dominated regime ($x < 0.5$). The field-like SOT from the same HHVR measurements is smaller than $\xi_{DL}^j$ for each $x$ and also varies with $x$ (Supplementary Fig. S7). We also measured $M_s$ and $\xi_{DL}^j$ of $Pt_{0.75}Ti_{0.25}$/ $Fe_{0.59}Tb_{0.41}$ as a function of temperature (we refer to this as the temperature series). Upon cooling from 350 K to 25 K, $M_s$ and $\xi_{DL}^j$ for the $Pt_{0.75}Ti_{0.25}$/$Fe_{0.59}Tb_{0.41}$ sample are tuned by >2 times (Fig. 2c) and by >7.5 times (Fig. 2d), respectively. The dramatic tuning of $\xi_{DL}^j$ by the $Fe_xTb_{1-x}$ composition and temperature is a striking observation because it suggests a strong dependence of SOTs on some bulk properties of the $Fe_xTb_{1-x}$, in contrast to $\xi_{DL}^j$ for heavy metal (HM)/3$d$ FM samples, which is insensitive to the type of the FM[33] and temperature[34,35]. Apparently, $\xi_{DL}^j$ for the $Pt_{0.75}Ti_{0.25}$/$Fe_xTb_{1-x}$ correlates closely with net magnetization (Fig. 2a, b) but not in a proportional or monotonic manner (Fig. 2e, f), suggesting a rather critical role of the net magnetization as well as another bulk effect (which, as we discuss below, is spin-orbit scattering) in the determination of $\xi_{DL}^j$.

### Robustness of the spin Hall ratio and the effective spin-mixing conductance
These strong variations cannot be explained by changes in either $\theta_{SH}$ or $T_{int}$ (Eq. (1)). First, $\theta_{SH}$ is a property of the $Pt_{0.75}Ti_{0.25}$ layer, not the $Fe_xTb_{1-x}$ layer. The $Pt_{0.75}Ti_{0.25}$ layer is made identically for all of the samples, and has a sufficiently large resistivity ($\rho_{xx} = 135\ \mu\Omega$ cm) such that its properties can hardly be affected significantly by either a neighboring layer or temperature. We have verified that $\xi_{DL}^j$ for a ferromagnetic $Pt_{0.75}Ti_{0.25}$ (5.6 nm)/FePt (8 nm) bilayer only has very weak temperature dependence (Supplementary Fig. S13), in good consistence with previous reports on other HM/3$d$ FM samples[34,35]. We have also measured negligible SOT signal from the 8 nm $Fe_xTb_{1-x}$ layers in control samples without a $Pt_{0.75}Ti_{0.25}$ layer (Supplementary Note 6), indicating that changes in our signals are not due to SOT arising from the $Fe_xTb_{1-x}$ bulk. Note that a bulk torque of a magnetic layer is strongly thickness dependent[36,37] and vanishes at small thicknesses of a few nm[36].

As for the possibility of changes in $T_{int}$, if we employ a drift-diffusion analysis[14–16], the effect on $T_{int}$ of spin backflow at the $Pt_{0.75}Ti_{0.25}$/$Fe_xTb_{1-x}$ interface should be proportional to the effective spin-mixing conductance ($G_{eff}^{\uparrow\downarrow}$) of the interface, i.e., $T_{int} \approx 2G_{eff}^{\uparrow\downarrow}/G_{PtTi}$[38],

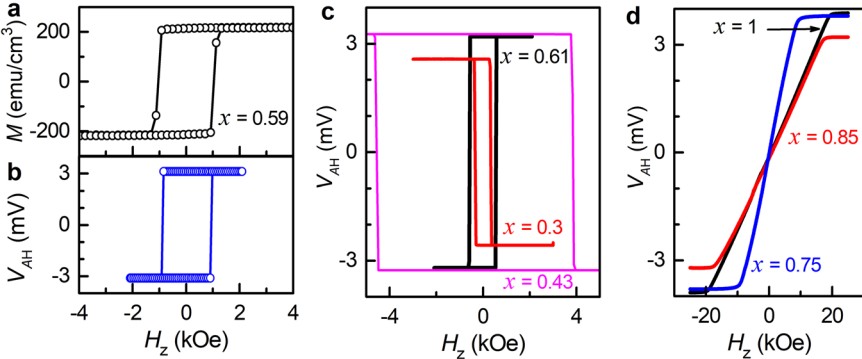

**Fig. 1 | Magnetization and anomalous Hall voltage hysteresis. a** Magnetization vs out-of-plane field ($H_z$) and **b** Anomalous Hall voltage ($V_{AH}$) vs $H_z$ for $Pt_{0.75}Ti_{0.25}$ (5.6 nm)/$Fe_{0.59}Tb_{0.41}$ (8 nm) under a sinusoidal electric field $E = 30$ kV/m, indicating strong perpendicular magnetic anisotropy and a high coercivity of ≈1 kOe. $V_{AH}$ vs $H_z$ for $Pt_{0.75}Ti_{0.25}$ (5.6 nm)/$Fe_xTb_{1-x}$ (8 nm) with **c** perpendicular ($x = 0.3, 0.43$, and 0.61) and **d** in-plane magnetic anisotropy ($x = 0.75, 0.85$, and 1).

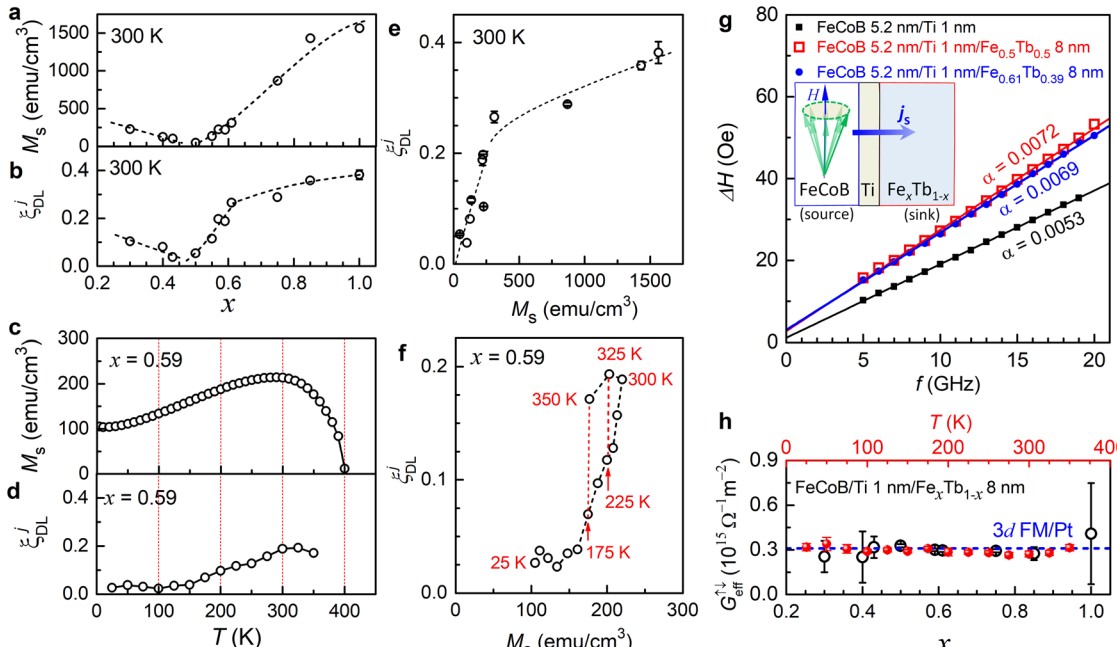

**Fig. 2 | Spin-orbit toque and spin-mixing conductance. a, b** Saturation magnetization ($M_s$) and Damping-like SOT efficiency per unit current density ($\xi^j_{DL}$) for $Pt_{0.75}Ti_{0.25}/Fe_xTb_{1-x}$ with different Fe concentration ($x$) at 300 K. **c, d** $M_s$ and $\xi^j_{DL}$ for the $Pt_{0.75}Ti_{0.25}/Fe_{0.59}Tb_{0.41}$ at different temperatures. **e** $\xi^j_{DL}$ vs $M_s$ for the $Pt_{0.75}Ti_{0.25}/Fe_xTb_{1-x}$ with different Fe concentration ($x$) at 300 K. **f** $\xi^j_{DL}$ vs $M_s$ for the $Pt_{0.75}Ti_{0.25}/Fe_{0.59}Tb_{0.41}$ at different temperatures. **g** Frequency dependence of ferromagnetic resonance linewidth ($\Delta H$) of the FeCoB layer in FeCoB (5.2 nm)/Ti (1 nm), FeCoB (5.2 nm)/Ti (1 nm)/$Fe_{0.5}Tb_{0.5}$ (8 nm), and FeCoB (5.2 nm)/Ti (1 nm)/$Fe_{0.61}Tb_{0.39}$ (8 nm) samples. The solid lines represent linear fits, the slopes of which yield the damping. In (**a–e**) some error bars are smaller than the data points. **h** $G^{\uparrow\downarrow}_{eff}$ of the FeCoB/Ti/$Fe_xTb_{1-x}$ interfaces measured from spin pumping into the $Fe_xTb_{1-x}$. The blue circles are for the composition series (300 K) and the red dots for the temperature series ($x = 0.59$). The blue dashed line represents $G^{\uparrow\downarrow}_{eff} = 0.31 \times 10^{15}$ $\Omega^{-1}$ $m^{-2}$ previously reported for typical Pt/$3d$ FM interfaces[33]. Error bars represent fitting uncertainty.

with $G_{PtTi} = 1/\lambda_s\rho_{xx} \approx 1.3 \times 10^5$ $\Omega^{-1}$ $m^{-1}$ being the spin conductance[18,39] of the $Pt_{0.75}Ti_{0.25}$. To quantify $G^{\uparrow\downarrow}_{eff}$, we measure the change in the damping ($\alpha$) of a precessing $Fe_{60}Co_{20}B_{20}$ (= FeCoB) layer due to the absorption of the FeCoB-emitted spin current at the $Fe_xTb_{1-x}$ interfaces (Fig. 2g and Supplementary Fig. S8). The samples used here had the structure FeCoB (5.2 nm)/Ti (1 nm) and FeCoB (5.2 nm)/Ti (1 nm)/$Fe_xTb_{1-x}$ (8 nm). Each of these samples is sputter-deposited on a 1 nm Ta seed layer and protected by capping with MgO (2 nm)/Ta (1 nm). The value of $\alpha$ for the FeCoB layers is determined from the linear dependence of the ferromagnetic resonance linewidth ($\Delta H$, half width at half maximum) on the frequency ($f$) using the relation $\Delta H = \Delta H_0 + 2\pi\alpha f/\gamma$, where $\Delta H_0$ is the inhomogeneous broadening of the linewidth and $\gamma$ the gyromagnetic ratio. The damping enhancement of the FeCoB layer induced by spin pumping into the 8 nm $Fe_xTb_{1-x}$ layers, $\Delta\alpha = \alpha_{FeCoB/Ti/FeTb} - \alpha_{FeCoB/Ti}$, can be related to $G^{\uparrow\downarrow}_{eff}$ as refs. [40–42]

$$\triangle\alpha = \gamma\hbar^2 G^{\uparrow\downarrow}_{eff}/2e^2M_{FeCoB}t_{FeCoB} \qquad (2)$$

where $t_{FeCoB} = 5.2$ nm and $M_{FeCoB} = 1255$ emu/$cm^3$ is the saturation magnetization of the FeCoB layer as measured by SQUID. The value of $\alpha = 0.0053$ for the bare FeCoB/Ti sample with no $Fe_xTb_{1-x}$ coincides closely with the intrinsic damping of FeCoB ($\approx 0.006$[33]), indicating that the damping in this system does not contain any significant contributions from interfacial two-magnon scattering or spin memory loss. As shown in Fig. 2h, $G^{\uparrow\downarrow}_{eff}$ of the FeCoB/Ti/$Fe_xTb_{1-x}$ interfaces is insensitive to temperature and the $Fe_xTb_{1-x}$ composition within experimental uncertainty, and has a value as high as that of typical $3d$ FM/Pt interfaces ($\approx 0.31 \times 10^{15}$ $\Omega^{-1}$ $m^{-2}$)[33]. This indicates that compensated $Fe_xTb_{1-x}$ alloys act as spin sinks that are just as good as $3d$ FMs and Pt, and that there is no enhancement in the amount of spin reflection and backflow due to magnetic compensation. In principle, changes in $T_{int}$ for SOT measurements could also arise from spin

memory loss induced by interfacial SOC[29,43,44], but this should be a minor effect for $T_{int}$ of our un-annealed $Pt_{0.75}Ti_{0.25}/Fe_xTb_{1-x}$ just as is the case of un-annealed Pt/Co[45]. As noted above, we also do not observe any enhancement in damping due to spin memory loss in the spin-pumping measurements. Note that the large and robust $G^{\uparrow\downarrow}_{eff}$ for electron-mediated spin transport at the metallic $Fe_xTb_{1-x}$ interface is in sharp contrast to that of ferrimagnetic insulator interfaces[46,47] where thermal magnons mediate the spin transport such that a very low magnetic moment density reduces $G^{\uparrow\downarrow}_{eff}$.

## Variation of SOT with the relative spin relaxation rates

Since we have ruled out any significant change in $\theta_{SH}$ or $T_{int}$ as contributing to the large variations we measure in $\xi^j_{DL}$ as a function of composition and temperature, these large variations must be due to physics that is not captured in the simple Eq. (1). We show below that spin relaxation induced by SOC in the bulk of the $Fe_xTb_{1-x}$ layer is the most likely physics that is neglected in Eq. (1). As schematically shown in Fig. 3a, a spin current, in general, can be relaxed in a magnetic layer through two competing mechanisms: exchange-interaction-induced angular momentum transfer from spin current to magnetization (with a relaxation rate $\tau_M^{-1}$ and a length scale of $\lambda_{dp}$) and bulk spin-orbit-scattering-induced loss of spin angular momentum to the lattice (with a relaxation rate $\tau_{so}^{-1}$ and a length scale of $\lambda_s$). Theory[24–27] and experiments[48,49] have suggested that, in fully or partially compensated magnetic systems, the rate of spin angular momentum transfer via exchange interaction can be greatly decreased because of cancellations between exchange fields of antiferromagnetically-aligned magnetic sub-lattices, resulting in long $\lambda_{dp}$ and low $\tau_M^{-1}$. Spin-orbit scattering is well known to result in spin relaxation in both magnetic and non-magnetic materials[22,23]. While relatively weak in light, highly conductive $3d$ FMs (e.g., $\lambda_s$ was measured to be 5–8 nm for Fe, Co, and CoFe at room temperature[22,23]), spin-orbit scattering becomes very

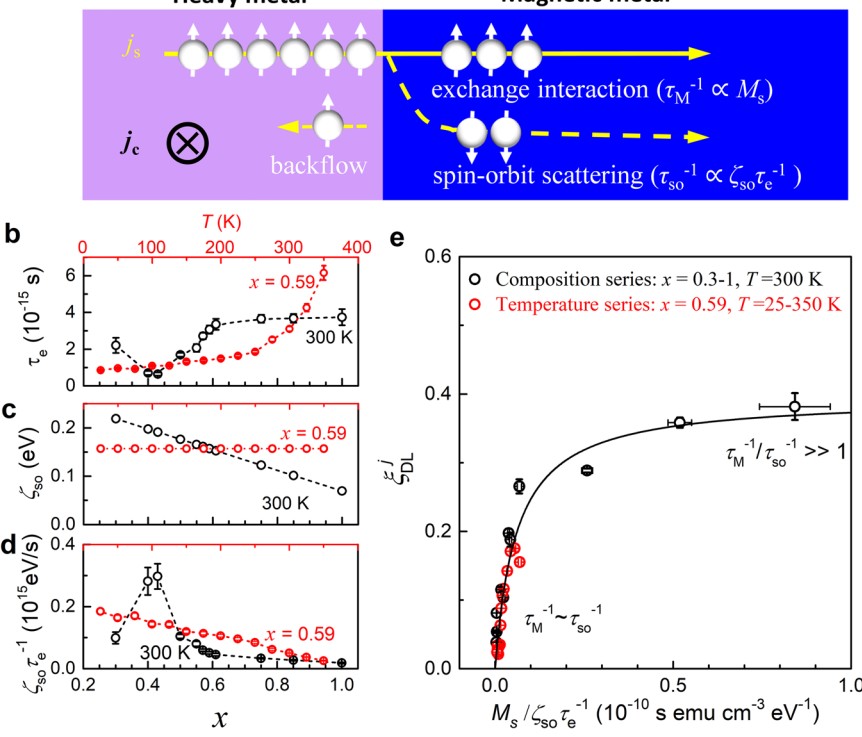

**Fig. 3 | Variation of SOT with the relative spin relaxation rates. a** Schematic of the spin relaxation processes that can influence the SOT, highlighting the competition between exchange interaction (with relaxation rate $\tau_M^{-1} \propto M_s$) and spin-orbit scattering ($\tau_{so}^{-1} \propto \zeta_{so}\tau_e^{-1}$). Only the spin current relaxed by exchange interaction contributes to SOTs. **b** Momentum scattering time ($\tau_e$), **c** Estimated SOC strength ($\zeta_{so}$), **d** $\zeta_{so}\tau_e^{-1}$, and **e** $\xi_{DL}^j$ of $Pt_{0.75}Ti_{0.25}/Fe_xTb_{1-x}$ vs $M_s/\zeta_{so}\tau_e^{-1}$ for the composition series ($x = 0.3-1$, $T = 300$ K, black circles) and for the temperature series ($x = 0.59$, $T = 25-300$ K, red circles). The solid curve in (**e**) represents the fit of the data to Eq. (3). Error bars represent fitting uncertainty.

strong in strong-SOC, resistive materials (e.g., dirty heavy metals[18] and rare-earth FIMs) and substantially reduces $\lambda_s$ and enhances $\tau_{so}^{-1}$. This makes it possible for spin currents in FIMs to relax partially or even primarily via spin-orbit scattering to the lattice, instead of applying a spin-transfer torque to the magnetization.

We can consider how the SOT should scale as a function of the ratio $\tau_M^{-1}/\tau_{so}^{-1}$. Quantitative measurements of these rates (e.g., from the dependence on layer thicknesses of spin valve or spin-pumping experiments) are quite challenging because the bulk properties of $Fe_xTb_{1-x}$[50] and other ferrimagnetic alloys[48,49] vary sensitively with the layer thicknesses[51–53] (e.g., the magnetic compensation, the bulk PMA, the orientation of the magnetic easy axis, and resistivity all change with thickness). Nonetheless, it is reasonable to expect $\tau_M^{-1} \propto M_s$ for such ferrimagnetic alloys considering the canceling effects of the exchange fields from the antiferromagnetically-aligned magnetic sub-lattices. For the spin-orbit scattering rate, the Elliot-Yafet mechanism[51–53] predicts $\tau_{so}^{-1} \propto \zeta_{so}\tau_e^{-1}$, where $\zeta_{so}$ is the bulk SOC strength and $\tau_e^{-1}$ is the momentum scattering rate. One can thus expect $\tau_M^{-1}/\tau_{so}^{-1} = kM_s/\zeta_{so}\tau_e^{-1}$, with $k$ being a constant. Since only the spin current relaxed through exchange interaction with magnetization contributes to SOTs, we propose that

$$\xi_{DL}^j \approx \xi_{DL,0}^j \tau_M^{-1}/(\tau_M^{-1} + \tau_{so}^{-1}) = \xi_{DL,0}^j \left(1 + (kM_s/\zeta_{so}\tau_e^{-1})^{-1}\right)^{-1} \quad (3)$$

where $\xi_{DL,0}^j$ is the value of $\xi_{DL}^j$ in the limit $\tau_M^{-1}/\tau_{so}^{-1} \gg 1$.

Figure 3b–d shows the estimated values of $\tau_e$, $\zeta_{so}$, and $\zeta_{so}\tau_e^{-1}$ for both our composition series ($x = 0.3$-1, $T = 300$ K, black symbols) and our temperature series ($x = 0.59$, $T = 25$–350 K, red symbols). Here, the value of $\tau_e$ is a rough estimate from the resistivity of the $Fe_xTb_{1-x}$ following Drude model $\rho_{FeTb} = m^*/ne^2\tau_e$, where $m^*$ is the effective mass of the carriers and $n$ is the carrier density measured from the ordinary

Hall coefficient ($R_{OH} = 1/ne$ for a single-band model[54,55], see Supplementary Note 5 for more details). $\tau_e$ of the $Fe_xTb_{1-x}$ varies by a factor of $\approx 6$ by composition and a factor of $\approx 7$ by temperature, suggesting a significant tuning of the Fermi surface properties. The increase of $\tau_e$ in the $Fe_xTb_{1-x}$ metal with raising temperature likely originates from electron-electron interaction[56] ($\rho_{FeTb} \propto T^{1/2}$, see Supplementary Fig. S14a) or magnetic Brillouin zone scattering (the periodic potentials due to antiferromagnetic alignment of the magnetic sublattices can produce an additional magnetic Brillouin zone, of smaller volume in $k$-space than the ordinary lattice potential, whose planes further incise and contort the Fermi surface[57]). Both electron-electron interaction and magnetic Brillouin zone lead to additional electron scattering manifesting as a resistivity upturn upon cooling.[56,57] The average SOC strength of the $Fe_xTb_{1-x}$ is estimated as $\zeta_{so} \approx x\zeta_{so,Fe} + (1-x)\zeta_{so,Tb}$ following the linear dependence on alloy composition of the bulk SOC[58] and the theoretical values[59] of $\zeta_{so,Fe} = 0.069$ eV for Fe and $\zeta_{so,Tb} = 0.283$ eV for Tb (while the actual values of $\zeta_{so,Fe}$ and $\zeta_{so,Tb}$ within the amorphous $Fe_xTb_{1-x}$, which are not trivial to obtain, may be slightly different from the theoretical ones, this estimation should, at least, provide a reasonable functional approximation for the expected dramatic variation of $\zeta_{so}$ as a function of the $Fe_xTb_{1-x}$ composition, from light Fe to Tb-rich $Fe_{0.3}Tb_{0.7}$). We estimate that $\zeta_{so}\tau_e^{-1}$ decreases by a factor of $>16$ as $x$ varies between 0.3 to 1 and by a factor of $>7$ as temperature increases from 25 K to 350 K (Fig. 3d). In Fig. 3e, we plot $\xi_{DL}^j$ as a function of $M_s/\zeta_{so}\tau_e^{-1}$ for both the composition series and the temperature series. As $M_s/\zeta_{so}\tau_e^{-1}$, or equivalently $\tau_M^{-1}/\tau_{so}^{-1}$, decreases, we find that $\xi_{DL}^j$ decreases first slowly and then rapidly towards a vanishing value. The variation of $\xi_{DL}^j$ with $M_s/\zeta_{so}\tau_e^{-1}$ can be fit very well by Eq. (3) with $\xi_{DL,0}^j = 0.395 \pm 0.022$ and $k = (1.66 \pm 0.23) \times 10^{11}\,s^{-1}\,emu^{-1}\,cm^3\,eV$. Note that the applicability of the Drude model, the approximated value of $m^*$, and the single-band model for the ordinary Hall effect for estimating $\tau_e^{-1}$ is not essential for our conclusion of the variation of

$\xi_{DL}^{j}$ with relative spin relaxation rates, since similar scaling in Fig. 3e is present even when simply plotting $\xi_{DL}^{j}$ as a function of $M_s/\zeta_{so}\rho_{xx}$ (Supplementary Fig. S11). In the above discussions, we ignored any effect of local distribution of magnetizations (also known as sperimagnetism[50]) because it, if present, may only indirectly affect the average spin relaxation rates of spin-magnetization exchange interaction and spin-orbit scattering via reducing the net magnetization and strengthening SOC-related momentum scattering of spin carriers, respectively.

Since the strong variation of $\xi_{DL}^{j}$ with relative spin relaxation rates we propose here is unlikely to be specific just to the HM/Fe$_x$Tb$_{1-x}$ system (as indicated by the widespread presence of spin-orbit scattering in various materials[22,23] and by the general fact that the SOT provided by a given spin-current generator is significantly weaker on FIMs than on FMs, see Table 1), we generalize Eq. (1) as

$$\xi_{DL}^{j} = T_{\text{int}}\theta_{\text{SH}}\tau_{\text{M}}^{-1}/\left(\tau_{\text{M}}^{-1}+\tau_{\text{so}}^{-1}\right). \qquad (4)$$

This generalized equation should apply to FMs, FIMs, and AFs that are metals or insulators. In magnetic insulators, an incident spin current carried by magnons transfers angular momentum to the magnetization via exchange interaction (with spin relaxation rate $\tau_{\text{M}}^{-1}$) and also to the lattice via spin-orbit scattering of spin carriers (with spin relaxation rate $\tau_{\text{so}}^{-1}$).

We note that our conclusions are contrary to some previous experiments, which reported $\xi_{DL}^{j}$ to remain constant[60–62] or even

**Table 1 | Comparison of $\xi_{DL}^{j}$ for FIMs and 3$d$ FMs in contact with spin current sources that have similar resistivities, thicknesses, and thus similar values of $\theta_{\text{SH}}$ and $T_{\text{int}}$**

| spin current source | $\xi_{DL}^{j}$ | | |
|---|---|---|---|
| | FIM | 3$d$ FM | ratio |
| Ta | −0.03 (CoTb)[5] | −0.12 (FeCoB)[10] | 4 |
| W | −0.04 (CoTb)[67] | −0.44 (FeCoB)[68] | 11 |
| Pt | 0.017 (CoTb)[5] | 0.15 (Co, FeCoB)[10] | 8.8 |
| Pt/NiO | 0.09 (CoTb)[6] | 0.6 (FeCoB)[35] | 6.7 |
| Pt$_{0.75}$Ti$_{0.25}$ | 0.05 (Fe$_{0.5}$Tb$_{0.5}$) | 0.38 (Fe) | 7.6 |
| Bi$_2$Se$_3$ | 0.13 (GdFeCo)[7] | 3.5 (NiFe)[69] | 27 |

diverge[63] near the magnetic compensation point of HM/CoTb or HM/CoFeGd bilayers. While it might be possible that $\tau_{\text{M}}^{-1}/\tau_{\text{so}}^{-1}$ is different in CoTb and CoFeGd compared to Fe$_x$Tb$_{1-x}$ (e.g., Gd has zero atomic orbital angular momentum[8] and thus considerably weaker SOC than Tb), we also question these previous conclusions for a variety of technical experimental reasons. In three of the previous experiments, the PMA of the FIM layer was weak and showed gradual magnetization hysteresis[62], non-parabolic first-harmonic signal in HHVR measurements[60,62,63], and/or non-linear second-harmonic signal in HHVR measurements[60,62] as a function of a small in-plane applied magnetic field. This indicates magnetization behavior outside of the simple macrospin model assumed in the HHVR analysis. The HHVR results in refs. [62,63] also applied "planar Hall correction", the latter, if significant, causes erroneous estimates of $\xi_{DL}^{j}$ (see refs. [30,64–66] for more discussions). References [60,61] reported substantial changes of sample properties before and after device patterning, resulting in large uncertainties in the estimation of $M_s$ and $\xi_{DL}^{j}$ for those samples. The loop-shift measurements in ref. [61] also applied large $dc$ current densities of ~$10^7$ A/cm$^2$, close to the switching current density, in the resistive Ta/Co$_x$Tb$_{1-x}$ samples, leading to considerable Joule heating that had significantly altered the temperature, $M_s$, and ferrimagnetic compensation points of those samples. The latter resulted in additional uncertainties in those loop-shift results of $\xi_{DL}^{j}$ and ultimately prevented resolving the variation of $\xi_{DL}^{j}$ with the bulk properties of FIMs (i.e., $\tau_{\text{M}}^{-1}/\tau_{\text{so}}^{-1}$) in that work.

## Scientific implications

Our finding of the critical role of the relative rates of spin-orbit-induced relaxation to the lattice versus spin transfer to the magnetization has important implications for this field and resolves outstanding puzzles in previous experiments. We first schematically demonstrate in Fig. 4a the effect of spin-orbit scattering on SOTs suggested by Eq. (4). Only in absence of spin-orbit scattering ($\tau_{\text{so}}^{-1}=0$), should the simple form of Eq. (1) apply such that $\xi_{DL}^{j}$ is independent of $\tau_{\text{M}}^{-1}$ and thus $M_s$ for a sufficiently thick magnetic layer with nonzero $M_s$. This is a good approximation only for magnetic materials with $\tau_{\text{M}}^{-1}/\tau_{\text{so}}^{-1} \gg 1$, e.g., 3$d$ FMs that have high $M_s$, low resistivity, and weak SOC. However, in the presence of non-negligible spin-orbit scattering ($\tau_{\text{so}}^{-1}>0$), $\xi_{DL}^{j}$ decreases more and more rapidly with reducing $\tau_{\text{M}}^{-1}/\tau_{\text{so}}^{-1}$ (with $\tau_{\text{M}}^{-1} \propto M_s$). This is generally the case of uncompensated "AF" domains ($M_s > 0$ emu/cm$^3$) and FIMs with strong SOC and large resistivities (e.g., Fe$_x$Tb$_{1-x}$ and Co$_x$Tb$_{1-x}$). However, $\xi_{DL}^{j}$ always diminishes at zero $\tau_{\text{M}}^{-1}$

**a** Spin-orbit torque efficiency

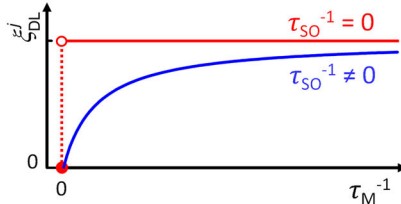

**b** Antiferromagnet ($M_s = 0$, $\xi_{DL}^{j} = 0$)

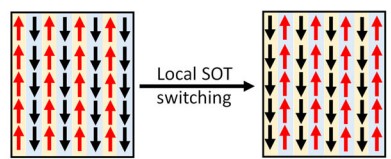

**c** Synthetic antiferromagnet ($M_s = 0$, $\xi_{DL}^{j} = 0$)

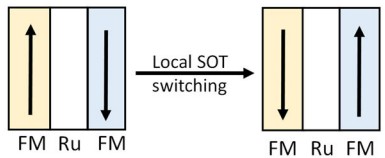

**d** Disordered ferrimagnet ($M_s > 0$, $\xi_{DL}^{j} > 0$)

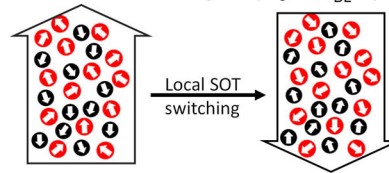

**Fig. 4 | Schematic depicts of the implications. a** Dependence on $\tau_{\text{M}}^{-1}$ ($\propto M_s$) of the efficiency of the damping-like spin-orbit torque ($\xi_{DL}^{j}$) exerted by a given spin current on a magnetic layer with zero spin-orbit scattering ($\tau_{\text{so}}^{-1} = 0$, red) and non-negligible spin-orbit scattering ($\tau_{\text{so}}^{-1} \neq 0$, blue), highlighting the critical role of spin-orbit scattering. "Potential" spin current switching of local atomic magnetizations by *local* spin-orbit torque in **b** perfectly compensated single-layer antiferromagnet, **c** perfectly compensated synthetic antiferromagnet, and **d** disordered ferrimagnet, with negligible barrier against switching (magnetic anisotropy, pinning, damping, etc.). In the case of perfectly compensated antiferromagnets, a spin current generates zero net spin-orbit torque and zero net magnetization change.

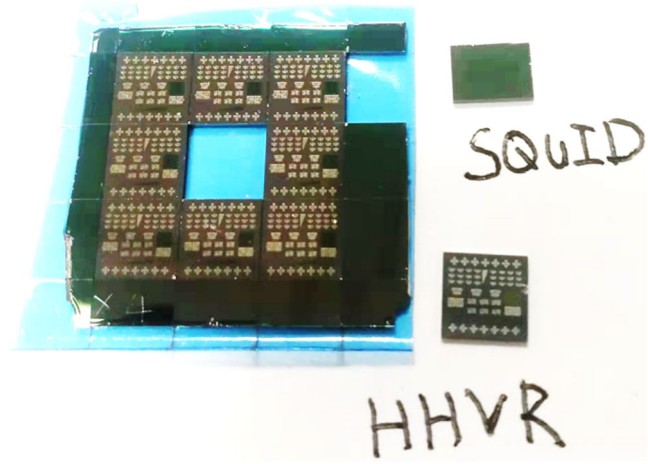

**Fig. 5 | Photo of a Pt$_{0.75}$Ti$_{0.25}$/Fe$_x$Tb$_{1-x}$ sample separated into pieces by a dicing saw.** The patterned pieces are used for HHVR measurements and un-patterned regions for SQUID measurements. The "SQUID" pieces underwent the same processing as the Hall bars during the device fabrication, providing good consistency between the electrical and magnetic properties of the samples in our analyses.

($M_s$ = 0 emu/cm$^3$), e.g., in perfectly compensated FIMs and AFMs and non-magnetic materials.

Therefore, SOTs will be reduced in any experiments which use magnetic free layers for which $\tau_M^{-1}$ is not much greater than $\tau_{so}^{-1}$ or not sufficiently large to allow complete spin relaxation within the layer thickness. The condition $\tau_M^{-1}/\tau_{so}^{-1} \ll 1$ resolves the puzzles why the measured $\xi_{DL}^j$ values for SOT acting on nearly-compensated FIMs are often several to over 20 times smaller than for corresponding measurements using 3$d$ FMs (see Table 1 for a few representative examples with spin current sources that have similar resistivities, thicknesses, and thus similar values of $\theta_{SH}$ and $T_{int}$)[5-7,10,35,67-69]. We have also verified from study of a large number of samples that $\tau_M^{-1}/(\tau_M^{-1} + \tau_{so}^{-1})$ is 0.58 for 5 nm Co$_{0.65}$Tb$_{0.35}$ layers at room temperature such that $\xi_{DL}^j$ for Pt-X/Co$_{0.65}$Tb$_{0.35}$ is only 58% of that of Pt-X/Co for given Pt-X (Pt-X being Pt-based alloys and multilayers)[70]. Spin-orbit scattering within the magnetic layer should, therefore, be minimized and the average exchange coupling maximized for efficient SOT devices.

Taking into account the relative spin relaxation rates also clarifies the ongoing debate as to whether AFs can be switched at all by SOTs[11-13]. The previous observation of current-driven switching of uncompensated magnetic domains or magnetizations embedded within AF hosts using transport or imaging methods[71-75] (which is not real switching of AF Néel vector) is naturally explained by the small but sizable SOTs in nearly but not fully-compensated systems, while the absence of macroscopic transport evidence of current switching of some more uniform HM/AF[11-13] is well consistent with the diminishment of SOTs in fully-compensated systems (Fig. 4a).

While in principle a spin current still has the potential to switch the local nonzero atomic magnetizations within a perfectly compensated FIMs, single-layer AFs, and synthetic AFs (e.g., FM/Ru/FM) by *local* SOT on each magnetic site (Fig. 4b–d), such switching is unfortunately very inefficient in overcoming the switching barriers of the samples (e.g., magnetic anisotropy, pinning field, damping, etc.). It is rather typical that the effective magnetic anisotropy fields and pinning fields (i.e., the coercivities in the magnetization hysteresis) of hard rare-earth transition-metal FIMs[50,60-62] and synthetic AFs[11] turn to diverge at the magnetization compensation point, making them unswitchable for a realistic spin current. Collective 180° reversal of the Néel vector of collinear AFs also requires net SOT to overcome the barriers against switching (e.g., magnetic anisotropy, pinning,

damping, etc.), which is challenging to achieve, particularly so in present of spin-orbit scattering. Therefore, fully-compensated FIMs and AFs themselves are most likely not an option as "Néel-vector-type" free layer of magnetic memory in terms of electrical writability and readability. We also note that field-like effective SOT field is too weak to reach the spin-flop field required for 90° rotation of Néel vector (typically of the order of 1–10$^3$ kOe)[76].

In summary, we have shown that the strength of SOTs depends critically on the ratio of rate of spin-orbit-induced spin relaxation within a magnetic layer relative to the rate of exchange-induced spin transfer to the magnetization. We find experimentally that SOT efficiencies decrease strongly upon approaching the magnetic compensation point in ferrimagnetic Fe$_x$Tb$_{1-x}$ due to a decrease in the rate of exchange-induced spin transfer on account of partial cancellation between the oppositely-directed exchange interactions from the magnetic sub-lattices. Near the compensation point, spin-orbit-induced spin relaxation dominates over spin transfer to the magnetization so that the measured SOT goes to zero. These results suggest the breakdown of the "interfacial torques" concept in FIMs and AFs. We find no indication of any dependence of the spin transparency of Fe$_x$Tb$_{1-x}$ interfaces on the degree of compensation. Our finding suggests that it will be essential to modify spin transport models that assumed an infinite $\tau_M^{-1}$ to include spin decoherence by spin-orbit scatting on an equal footing with dephasing by the exchange interaction. This work provides not only a unified understanding of the very different efficiencies of SOTs that have been reported in the literature for FMs, FIMs, and AFs, but also insight about how the different sources of spin relaxation should be optimized in the design of FIMs and AFs for spintronic technologies[8,9].

## Methods
### Sample fabrication
Samples for this study includes Pt$_{0.75}$Ti$_{0.25}$ (5.6 nm)/Fe$_x$Tb$_{1-x}$ (8 nm) bilayers, FeCoB (5.2 nm)/Ti (1 nm)/Fe$_x$Tb$_{1-x}$ (8 nm), FeCoB (5.2 nm)/Ti (1 nm), Pt$_{0.75}$Ti$_{0.25}$ (5.6 nm)/FePt (8 nm), and Fe$_x$Tb$_{1-x}$ (8 nm), with x being the Fe volumetric concentration (x = 0.3−1). Each sample was sputter-deposited on an oxidized Si substrate with a 1-nm Ta seed layer, and protected by a 2 nm MgO and a 1.5 nm Ta layer that was oxidized upon exposure to atmosphere. The FeCoB (5.2 nm)/Ti (1 nm)/Fe$_x$Tb$_{1-x}$ (8 nm) and FeCoB (5.2 nm)/Ti (1 nm) samples were diced into pieces for ferromagnetic resonance measurements. The other samples (2 × 2 cm in area) were partly patterned into 5 × 60 μm$^2$ Hall bars by photolithography and ion milling with a water-cooled stage. After processing, the samples were separated into pieces by a dicing saw (Fig. 5). The patterned pieces were used for harmonic Hall voltage response (HHVR) measurements, and un-patterned regions of the films for magnetization characterizations using a superconducting quantum interference device (SQUID). The "SQUID" pieces underwent the same processing as the Hall bars during the device fabrications, providing good consistency between the electrical and magnetic properties of the samples in our analyses.

### Measurements
The saturation magnetization of each sample was measured by a Quantum Design SQUID as a function of magnetic field and temperature. The SOTs were measured using harmonic Hall voltage response (HHVR) measurements, during which a Signal Recovery DSP Lock-in Amplifier (Model 7625) or a Stanford Research System Lock-in Amplifier (Model SR830) was used to source a sinusoidal electric field $E$ (typically 30 kV/m) onto the Hall bars and to detect the first and second harmonic Hall voltage responses. The magnetic damping was measured using flip-chip ferromagnetic resonance with a radio frequency signal generator, two Stanford Research System Lock-in Amplifiers (Model SR830), and a sweeping in-plane magnetic field.

**Reporting summary**

Further information on research design is available in the Nature Portfolio Reporting Summary linked to this article.

## Data availability

Source data are provided with this paper.

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

## Acknowledgements

We thank Dahai Wei, Xin Lin, Qianbiao Liu for help with sample deposition. We also thank Tianxiang Nan, Yanan Chai, Wei Han, Liangliang Guo for help with ferromagnetic resonance measurements. This work was supported in part the National Key Research and Development Program of China (Grant No. 2022YFA1200094), in part by the National Natural Science Foundation of China (Grant No. 12274405), in part by the Strategic Priority Research Program of the Chinese Academy of Sciences (XDB44000000), in part by the NSF MRSEC program (DMR-1719875) through the Cornell Center for Materials Research, and in part by the NSF (NNCI-2025233) through the Cornell Nanofabrication Facility/National Nanotechnology Coordinated Infrastructure.

## Author contributions

L.Z. conceived the study and performed the experiments. L.Z. and D.C.R. analyzed the data and wrote the paper.

## Competing interests

The authors declare no competing interests.
