## [Peer Review File · Nature Communications]

Reviewers' Comments:

Reviewer #1:

Remarks to the Author:

This work examines the mechanism of the damping-like spin-orbit torque in ferrimagnetic FeTb alloys interfaced with PtTi (spin-Hall source metal). The authors show a correlation between the SOT efficiency (ξ^j_{DL} , measured using the second harmonic Hall voltage, HHVR, method) and the net saturation magnetization M_s of the ferrimagnetic alloy. Both compositional and temperature dependent measurements of the efficiency are reported here. Notably, the measured efficiency declines to a small value near the magnetic compensation point of FeTb, where M_s becomes small. The authors attribute this trend to the competing relaxation mechanisms (dephasing and spin-orbit scattering) of the spin current injected into the FeTb layer. In particular, as FeTb approaches magnetic compensation, the dephasing of the spin current is slowed down, such that the spin current is absorbed (relaxes) in FeTb via spin-orbit scattering. That is, the authors claim that near the compensation point, more of the spin angular momentum is lost directly to the lattice rather than exerting a torque on the magnetic order.

I believe the perspective the authors present here is interesting and unique. There have been not many studies that examine the fate of a transverse spin current injected in the "bulk" of the magnetic medium. More importantly, the authors raise an interesting fundamental question regarding the roles of (1) dephasing and (2) spin-orbit scattering in the relaxation of transverse spin currents in ferrimagnetic alloys, key materials that have gained considerable attention from the spintronics community recently. Overall, this study may stimulate a new direction in understanding and engineering spin torques in magnetic heterostructures.

While the authors' interpretation (i.e., competition between dephasing and scattering) is fundamentally interesting and has important practical implications, I would like the authors to consider and address the following points in their revised manuscript.

1) I think the authors overestimate both the spin dephasing length and the spin diffusion (spin flip) length in FeTb.

I have difficulty believing that the spin dephasing length is ~ 10 nm in FeTb as the authors state on Page 7. The dephasing length in FeTb is likely only $\sim 4-5$ nm at most (as reported by Phys. Rev. B 103, 024443 (2021) for CoGd). The very long dephasing length of ~ 10 nm probably requires a clean system (with limited momentum scattering) with layer-by-layer collinear antiferromagnetic order (that can counteract dephasing). Note that even spin-conserving momentum scattering can greatly decrease the coherence length scale of the spin current [see for example, Saidaoui & Manchon, Phys. Rev. B. 89, 174430 (2014)]. With that in mind, there is likely enough momentum scattering in structurally disordered FeTb alloys here, resulting in dephasing length $\ll 10$ nm. It is also highly doubtful that FeTb resembles a layer-by-layer antiferromagnet; rather FeTb is probably a sperimagnet with noncollinear magnetic order. In that regard, the dephasing length in FeTb with noncollinear magnetic order may be even a bit shorter than $\sim 4-5$ nm in CoGd (with presumably closer to collinear magnetic order).

I would think that the upper bound of the spin diffusion length in FeTb is much shorter than 8 nm. A recent experimental study [Zahnd et al. Phys Rev B 98, 174414 (2018)] shows a spin diffusion length $\sim 5-6$ nm for NiFe and CoFe alloys. It is probably considerably shorter for FeTb, in which Tb leads to greater spin-orbit scattering.

Of course, the shorter dephasing and diffusion lengths would not preclude the scenario that the authors claim – that spin-orbit scattering relaxation becomes comparable to the spin transfer process with partial (or even slight?) rephasing of spin current near magnetic compensation and sufficiently strong spin-orbit scattering from FeTb. I am just concerned that the length scales (~ 10 nm) that the authors reference here are quite exaggerated. (In fact, the likely shorter diffusion length makes the authors' claim more credible.)

2) I recognize that quantifying the "spin-orbit coupling (SOC) strength" is not trivial for the structurally disordered amorphous alloys studied here, so that referencing the theoretical calculation results for simple square lattices (Ref 60) is perhaps the only viable way. Still, I believe that the authors should be more explicit about the caveat in the quantitative estimation of the SOC strength. There is also probably a caveat for the estimation of the momentum scattering rate

(simple Drude model, with electron effective mass equal to the mass of the free electron). In other words, overall, the estimates here are more qualitative.

3) The authors' results (e.g., Figure 2 (a-d)) show that the SOT efficiency roughly correlates with the net magnetization. Thus, perhaps another possible interpretation of the results is that the SOT simply acts on the net magnetization, i.e., vanishing as the FeTb alloy becomes nearly compensated. With this simple alternative interpretation, there would no need to invoke the role of spin-orbit scattering to explain the observation. The authors should explicitly comment on this alternative interpretation (spin transfer acting on the next magnetization).

4) As the authors acknowledge, there have been many reports of SOTs in ferrimagnetic alloys showing results that are very different from the author's present study. (In fact, I might recommend adding one more to the referenced papers here: Je et al. Appl. Phys. Lett. 112, 062401 (2018).) To the best of my knowledge, the authors' experimental study is the only one so far that shows the SOT efficiency *decreases* (seemingly vanishes) as the alloy approaches magnetic compensation. There is one report that shows arguably something similar – i.e., inability to switch by SOT a compensated synthetic ferrimagnet [Appl. Phys. Lett. 117, 172403 (2020)] – but perhaps the physics at work is a bit different from ferrimagnetic alloys. There still appears to be quite a bit of controversy surrounding the HHVR method (e.g., what to do about the planar Hall voltage contribution), as the authors acknowledge on Page 7. I think the authors' work could be strengthened if a similar trend can be seen with another SOT measurement method, such as the hysteresis loop shift method [Phys. Rev. B 93, 144409 (2016)], used in other prior studies on PMA ferrimagnetic alloys. Alternatively, the authors could explain why HHVR (despite the ongoing controversy) is still more reliable than the hysteresis loop shift method.

5) Related to the point above, how reliable are the three HHVR-derived SOT efficiency data points for the in-plane anisotropy samples? Can they be compared on the same ground as the data for the PMA samples?

6) The authors' claim implies that it is very difficult, or perhaps impossible, to switch metals with compensated antiferromagnetic order. This would be very bad news for the field of antiferromagnetic spintronics. I am not sure whether the authors want to convey the bad news more explicitly (than what they already say toward the bottom of Page 7), but I feel that this is another big far-reaching implication of this present study.

7) Related to the point above, Refs 12 and 13 are on insulating antiferromagnets. If space permits, it may be worth pointing out that the mechanisms of transverse spin current transport and relaxation are likely different between metallic and insulating ferri/antiferromagnets.

Reviewer #2:

Remarks to the Author:

This manuscript reports the spin-orbit-torque (SOT) measurement in ferrimagnetic TbFe. There are many reports on the SOT in ferrimagnets, but some discrepancies appear among the reports thus far. This is because the ferrimagnets, in particular the rare earth-transition metal (RE-TM) ferrimagnets, are very sensitive to the composition and temperature, and even to the oxidation. Therefore, it is highly desirable to get a comprehensive understanding on the SOT in ferrimagnets.

This work presents systematic studies on the SOT in ferrimagnets, which can be accomplished by changing the composition and temperature in a controlled manner. Based on their observations, the authors propose a more complete description about the SOT in ferrimagnets. They claim that the SOT is proportional to the (exchange-induced angular momentum transfer rate/spin-orbit-scattering loss rate). More simply speaking, the SOT is proportional to the magnetization (M_s), and thus the SOT would be negligible in antiferromagnet. Furthermore, the SOT is inversely proportional to momentum scattering rate, which suggests that the high SOC material is disadvantageous for SOT switching.

This kind of result can certainly advance our understanding by providing a deep physical insight,

and therefore will lead a community. At the same time, however, it could mislead the community if the conclusion is made based on weak evidences.

After reading the manuscript, I am still not fully convinced that the authors claimed in this manuscript because of the following reasons.

1. There are many reports about the existence of SOT in antiferromagnetic systems (e.g., DuttaGupta et al., Nat. Commun. 11, 5715 (2020), Kim et al., PRB 104, 054406 (2021), Tsai et al., AIP advances 11, 045110 (2021), etc). Do these results compatible with the authors' claim?
2. In the experiment, the magnetization varies orders of magnitude (from 1500 emu/cc to nearly zero), but the spin-orbit scattering changes at most factor 10. It seems that the results can be explained solely by the variation of magnetization. Why do they think the bulk spin-orbit-scattering is a key to understand their results?
3. I feel that the analysis on the spin-orbit-scattering shown in Fig. 3 and sec. 6 of SI is a bit clumsy. More rigorous analysis is required by considering the followings.
 - 3-1. They used simple Drude model to get the transport parameters. Do they confident that the Drude model can be applicable to the RE-TM ferrimagnets which are amorphous and highly resistive compared to the normal metals?
 - 3-2. They assumed that the electron effective mass was equal to the free electron mass. However, the electron effective mass in solid is different to the free electron mass. Furthermore, the effective mass in solid is strongly dependent on the band structure as well as on the temperature. As they changed the composition and temperature of TbFe, the effective mass should have some variation.
 - 3-3. More detailed description about how they subtracted the ordinary Hall effect of PtTi is required.
 - 3-4. In Fig. S10(b), the ordinary Hall resistance seems to rapidly increase at higher temperature. How do they explain this? Is it possible to come from the paramagnetic effect with approaching Curie temperature?
 - 3-5. It is generally understood that the carrier density strongly depends on the band structure, which means that the carrier density is more sensitive to the composition variation rather than the temperature change. However, Fig. S10(d) shows that the amount of the variation of carrier density is almost same for both cases.
4. TbFe is known as a "sperimagnet", in which the Tb and Fe moments are spread out and form a cone angle due to the strong anisotropy of Tb and weak exchange of Fe (Hebler et al., Fron. Mater. 3, 8 (2016) doi: 10.3389/fmats.2016.00008). Recent report showed that the sperimagnetic properties caused exotic phenomena (Park et al., Nat. Commun. 13, 5530 (2022)). As the sperimagnet has a dispersed spin structure, it can also affect to the spin scattering. Can the authors exclude this possibility?
5. The magnetic moment of Fe arises from the 3d orbital electron, while that of Tb is mainly determined by the 4-f orbital electron which lies far lower than the Fermi level. In this manuscript, the authors considered the Fe and Tb moments equally when considering the spin scattering. What is microscopic background in which conduction electrons interact with 4-f electrons through the exchange interaction and SOC?

Based on above concerning, I think the present version of the manuscript is premature to be published.

<Minor comments>

1. p. 3, below the Fig. 2: "thermoelectric effects (see details in Sec. 1 in the Supplementary Materials)." Sec. 1 should be Sec. 3
2. I recommend to denote the current density that they used in their experiments.
3. In Fig. S5(a), mark the origin (zero field point) on the graph
4. In Fig. S5(b)-(f), why does the 2nd harmonic signal have some offset for zero field?
5. Fig. S7(a) is inconsistent with other data. According to the authors' claim, the damping-like effective field should be proportional to the magnetization. However, it seems that the data in Fig.

S7(a) is somewhat opposite.

Response Letter

We are grateful to both of the referees for their constructive, helpful comments and suggestions. Following the referees' suggestions, we have revised our manuscript and Supplementary Information. We believe these revisions have made our work more compelling. We hope the referees find that we have adequately addressed all the comments of the referees. Below we provide point-by-point responses to the comments of the referees (Page 1-7 for Referee #1 and Page 8-14 for referee #2).

Reviewer #1:

Comment A-1: *This work examines the mechanism of the damping-like spin-orbit torque in ferrimagnetic FeTb alloys interfaced with PtTi (spin-Hall source metal). The authors show a correlation between the SOT efficiency (ξ^j_{DL} , measured using the second harmonic Hall voltage, HHVR, method) and the net saturation magnetization M_s of the ferrimagnetic alloy. Both compositional and temperature dependent measurements of the efficiency are reported here. Notably, the measured efficiency declines to a small value near the magnetic compensation point of FeTb, where M_s becomes small. The authors attribute this trend to the competing relaxation mechanisms (dephasing and spin-orbit scattering) of the spin current injected into the FeTb layer. In particular, as FeTb approaches magnetic compensation, the dephasing of the spin current is slowed down, such that the spin current is absorbed (relaxes) in FeTb via spin-orbit scattering. That is, the authors claim that near the compensation point, more of the spin angular momentum is lost directly to the lattice rather than exerting a torque on the magnetic order.*

I believe the perspective the authors present here is interesting and unique. There have been not many studies that examine the fate of a transverse spin current injected in the “bulk” of the magnetic medium. More importantly, the authors raise an interesting fundamental question regarding the roles of (1) dephasing and (2) spin-orbit scattering in the relaxation of transverse spin currents in ferrimagnetic alloys, key materials that have gained considerable attention from the spintronics community recently. Overall, this study may stimulate a new direction in understanding and engineering spin torques in magnetic heterostructures.

Response: We are grateful to the referee for the very insightful comment on the fundamental importance of our work and also for the helpful suggestions.

Comment A-2: *While the authors' interpretation (i.e., competition between dephasing and scattering) is fundamentally interesting and has important practical implications, I would like the authors to consider and address the following points in their revised manuscript.*

1) I think the authors overestimate both the spin dephasing length and the spin diffusion (spin flip) length in FeTb. I have difficulty believing that the spin dephasing length is ~10 nm in FeTb as the authors state on Page 7. The dephasing length in FeTb is likely only ~4-5 nm at most (as reported by Phys. Rev. B 103, 024443 (2021) for CoGd). The very long dephasing length of ~10 nm probably requires a clean system (with limited momentum scattering) with layer-by-layer collinear antiferromagnetic order (that can counteract dephasing). Note that even spin-conserving momentum scattering can greatly decrease the coherence length scale of the spin current [see for example, Saidaoui & Manchon, Phys. Rev. B. 89, 174430 (2014)]. With that in mind, there is likely enough momentum scattering in structurally disordered FeTb alloys here, resulting in dephasing length $\ll 10$ nm. It is also highly doubtful that FeTb resembles a layer-by-layer antiferromagnet; rather FeTb is probably a sperimagnet with noncollinear magnetic order. In that regard, the dephasing length in FeTb with noncollinear magnetic order may be even a bit shorter than ~4-5 nm in CoGd (with presumably closer to collinear magnetic order).

Response: This seems to be a misunderstanding. The spin dephasing length of “**10 nm**” the referee concerned was the estimate for **CoTb** from the reference [Nat. Mater. 18, 29 (2019), originally Ref. 53], not for **FeTb**. Our original comment was “**Reference 68 studied CoTb layers with thicknesses (1.7-2.6 nm) much thinner than the likely spin dephasing length ($\approx 10 \text{ nm}^{53}$) so that the escape rate from the film was likely faster than either τ_M^{-1} or τ_{so}^{-1} .**” This comment has been removed from our revised manuscript because we now consider that spin current may get relaxed within the small thickness (1.7-2.6 nm) anyway due to the very short diffusion length near the compensation of the CoTb samples in the literature works.

We emphasize that we only need the ratio of $\tau_M^{-1}/\tau_{so}^{-1}$ in our analysis and have never attempted estimated the exact lengths and rates for spin dephasing and spin-orbit scattering in the $\text{Fe}_x\text{Tb}_{1-x}$, please refer to pages 5 and 6 of our manuscript, “**Quantitative measurements of these rates (e.g., from the dependence on layer thicknesses of spin valve or spin-pumping experiments) are quite challenging because the bulk properties of $\text{Fe}_x\text{Tb}_{1-x}^{50}$ and other ferrimagnetic alloys^{48,49} vary sensitively with the layer thicknesses⁵¹⁻⁵³ (e.g., the magnetic compensation, the bulk PMA, the orientation of the magnetic easy axis, and resistivity all change with thickness). Nonetheless, it is reasonable to expect $\tau_M^{-1} \propto M_s$ for such ferrimagnetic alloys considering the cancelling effects of the exchange fields from the antiferromagnetically-aligned magnetic sub-lattices. For the spin-orbit scattering rate, the Elliot-Yafet mechanism⁵¹⁻⁵³ predicts $\tau_{so}^{-1} \propto \zeta_{so}\tau_e^{-1}$, where ζ_{so} is the bulk SOC strength and τ_e^{-1} is the momentum scattering rate. One can thus expect $\tau_M^{-1}/\tau_{so}^{-1} = kM_s/\zeta_{so}\tau_e^{-1}$, with k being a constant.**”

Comment A-3: *I would think that the upper bound of the spin diffusion length in FeTb is much shorter than 8 nm. A recent experimental study [Zahnd et al. Phys Rev B 98, 174414 (2018)] shows a spin diffusion length $\sim 5\text{-}6 \text{ nm}$ for NiFe and CoFe alloys. It is probably considerably shorter for FeTb, in which Tb leads to greater spin-orbit scattering. Of course, the shorter dephasing and diffusion lengths would not preclude the scenario that the authors claim – that spin-orbit scattering relaxation becomes comparable to the spin transfer process with partial (or even slight?) rephasing of spin current near magnetic compensation and sufficiently strong spin-orbit scattering from FeTb. I am just concerned that the length scales ($\sim 10 \text{ nm}$) that the authors reference here are quite exaggerated. (In fact, the likely shorter diffusion length makes the authors’ claim more credible.)*

Response: The upper bound of spin diffusion length (λ_s) of the $\text{Fe}_x\text{Tb}_{1-x}$ ($x=0.3\text{-}1$) samples is actually λ_s of Fe. Previous experiments have reported $\lambda_s \approx 8 \text{ nm}$ for 3d ferromagnets at room temperature. For instance, Refence 22 of our manuscript reported $8.5 \pm 1.5 \text{ nm}$ for Fe (Table 3 in that reference) and ref. 23 of our manuscript, which is actually the paper suggested by the referee [Zahnd et al. Phys Rev B 98, 174414 (2018)], reported $\approx 8 \text{ nm}$ for Co (300 K) and $6.2\text{-}8.3$ for CoFe (10-300 K). Please see the Figure below for the λ_s results from ref. 23. In any case, the exact value of this upper bound of λ_s is not critical for our analysis because no exact value of λ_s is used to estimate anything in our discussions.

TABLE I. Spin transport parameters extracted from the spin-absorption experiment for each material, at both room and low (10 K) temperature. The resistivities are measured using a Van der Pauw method.

	ρ (300 K) $\mu\Omega \text{ cm}$	P_F (300 K)	λ (300 K) nm	ρ (10 K) $\mu\Omega \text{ cm}$	P_F (10 K)	λ (10 K) nm
CoFe	20 ± 1.3	$0.48^{+0.0}_{-0.02}$	$6.2^{+0.3}_{-0.7}$	15 ± 0.9	$0.48^{+0.03}_{-0.01}$	$8.3^{+0.7}_{-1.8}$
NiFe	30 ± 3	$0.22^{+0.05}_{-0.06}$	$5.2^{+1.8}_{-0.9}$	22 ± 1.2	$0.40^{+0.1}_{-0.03}$	$5.8^{+0.2}_{-1.8}$
Co	25 ± 2.4	$0.17^{+0.08}_{-0.02}$	$7.7^{+1.8}_{-2.2}$	15 ± 1.6	$0.18^{+0.09}_{-0.03}$	$12.5^{+3.5}_{-3.7}$

We have added on Page 5 of our revised manuscript that “**While relatively weak in light, highly conductive 3d FMs (e.g., λ_s was measured to be 5-8 nm for Fe, Co, and CoFe at room temperature^{22,23}), spin-orbit**

scattering becomes very strong in strong-SOC, resistive materials (e.g., dirty heavy metals¹⁸ and rare-earth FIMs) and substantially reduces λ_s and enhances τ_{so}^{-1} ” and deleted previous comment that “the spin diffusion length is expected to be ≤ 8 nm at temperatures in this study^{22,23}.”

Comment A-4: 2) *I recognize that quantifying the “spin-orbit coupling (SOC) strength” is not trivial for the structurally disordered amorphous alloys studied here, so that referencing the theoretical calculation results for simple square lattices (Ref 60) is perhaps the only viable way. Still, I believe that the authors should be more explicit about the caveat in the quantitative estimation of the SOC strength. There is also probably a caveat for the estimation of the momentum scattering rate (simple Drude model, with electron effective mass equal to the mass of the free electron). In other words, overall, the estimates here are more qualitative.*

Response: Thanks for the very careful reading of our manuscript.

Following the suggestions of the referee, we have added on Page 7 of our revised manuscript that “(while the actual values of $\zeta_{so,Fe}$ and $\zeta_{so,Tb}$ within the amorphous Fe_xTb_{1-x} , which are not trivial to obtain, may be slightly different from the theoretical ones, this estimation should, at least, provide a reasonable functional approximation for the expected dramatic variation of ζ_{so} as a function of the Fe_xTb_{1-x} composition, from light Fe to Tb-rich $Fe_{0.3}Tb_{0.7}$)”

On page 7 of our revised manuscript, we have also note that “the value of τ_e is a rough estimate from the resistivity of the Fe_xTb_{1-x} following the Drude model... We note that the applicability of the Drude model, the approximated value of m^* , and the single-band model for the ordinary Hall effect for estimating τ_e^{-1} is not essential for our conclusion of the strong variation of ξ_{DL}^j with relative spin relaxation rates, since similar scaling in Fig. 3e is present even when simply plotting ξ_{DL}^j as a function of $M_s/\zeta_{so}\rho_{xx}$ (Supplementary Fig. S11).”

At the beginning of Supplementary Note 5, we had noted that “We make a rough estimate of the momentum-scattering time (τ_e) of the Fe_xTb_{1-x} from the resistivity....”

Comment A-5: 3) *The authors’ results (e.g., Figure 2 (a-d)) show that the SOT efficiency roughly correlates with the net magnetization. Thus, perhaps another possible interpretation of the results is that the SOT simply acts on the net magnetization, i.e., vanishing as the FeTb alloy becomes nearly compensated. With this simple alternative interpretation, there would no need to invoke the role of spin-orbit scattering to explain the observation. The authors should explicitly comment on this alternative interpretation (spin transfer acting on the next magnetization).*

Response: While that could be the impression at the first glance at Fig. 2a,b, the precise conclusion is that the experimental observation cannot be explained solely based on the net magnetization.

Stimulated by the referee’s comment, we have added in our revised manuscript that:

Page 4: “Apparently, ξ_{DL}^j correlates closely with net magnetization (Fig. 2a,b) but not in a proportion or monotonic manner (Fig. 2e,f), suggesting a rather critical role of the net magnetization and another bulk effect (which, as we discuss later, is spin-orbit scattering) in the determination of ξ_{DL}^j .”

Page 5: “Spin-orbit scattering is well known to result in spin relaxation in both magnetic and nonmagnetic materials^{22,23}.”

On Page 8,9 of our revised manuscript, we have also added Fig. 4a (attached below) and an extended discussion on the critical role of the spin-orbit scattering. In the case of zero spin-orbit scattering, spin current entering the magnetic layer would be fully transfer to the magnetization such that the torque would NOT vary with the net magnetization as soon as the latter is not exactly zero and the magnetic material is sufficiently thick. **This is not consistent with our observation or with the common sense that spin current also relaxes in non-magnetic materials with a scale length called spin diffusion length** (e.g., 1-2 nm for typical sputter-deposited Pt).

We note that ref. 47 observed a thickness dependence of SOTs in a ferrimagnetic insulator related to thickness dependent magnetization. **In our view**, there were three mechanisms at work in that case:(1) the *thermal magnon*-mediated spin mixing conductance of the magnetic insulators may be sensitive to the very low density of magnetic moment at the interface (which is possible, as indicated by theory in ref. 46); (2) the thickness variation is important as it is not much greater than the spin relaxation length which is long due to the very low magnetization and damping; (3) the increase of magnetization with thickness leads to enhancement of $\tau_M^{-1}/\tau_{so}^{-1}$ and thus torque with thickness. The first two mechanisms are clearly not at work in the case for our metallic FeTb samples. As we have determined on page 4 of the manuscript, the electron-mediated spin-mixing conductance of our FeTb interface is robust to composition and temperature, and the thickness of 8 nm is greater than the spin relaxation length set by spin dephasing and spin-orbit scattering together.

Comment A-6: 4) As the authors acknowledge, there have been many reports of SOTs in ferrimagnetic alloys showing results that are very different from the author’s present study. (In fact, I might recommend adding one more to the referenced papers here: Je et al. Appl. Phys. Lett. 112, 062401 (2018).) To the best of my knowledge, the authors’ experimental study is the only one so far that shows the SOT efficiency **decreases** (seemingly vanishes) as the alloy approaches magnetic compensation. There is one report that shows arguably something similar, i.e., inability to switch by SOT a compensated synthetic ferrimagnet [Appl. Phys. Lett. 117, 172403 (2020)]—but perhaps the physics at work is a bit different from ferrimagnetic alloys.

Response: Thanks for suggestion. The paper Appl. Phys. Lett. 112, 062401 (2018) did not indicate any “decrease” or disappearance of the torque near the magnetic compensation, instead it reported very scattered **POSITIVE** spin Hall angles of 0.1-0.3 with very large uncertainties for W/CoTb (Fig. 4c in that paper is attached below). The latter is not consistent with the well-known fact that the spin Hall angle of W is **negative**. That work seemed to contain critical errors likely due to careless analysis. Therefore, that work is not representative and we have decided not to use the space of our paper to comment on that work.

The second paper Appl. Phys. Lett. 117, 172403 (2020) is actually Ref. 11 in our manuscript. On page 9 of our revised manuscript, we have commented that “the absence of macroscopic transport evidence of current switching of some more uniform HM/AF¹¹⁻¹³ is well consistent with the diminishment of SOTs in fully-compensated systems (Fig. 4a).It is rather typical that the effective magnetic anisotropy fields and pinning fields (i.e., the coercivities in the magnetization hysteresis) of hard rare-earth transition-metal FIMs^{58,63-65} and synthetic AFs¹¹ turn to diverge at the magnetization compensation point, making them unswitchable for a realistic spin current.”

Comment A-7: *There still appears to be quite a bit of controversy surrounding the HHVR method (e.g., what to do about the planar Hall voltage contribution), as the authors acknowledge on Page 7. I think the authors' work could be strengthened if a similar trend can be seen with another SOT measurement method, such as the hysteresis loop shift method [Phys. Rev. B 93, 144409 (2016)], used in other prior studies on PMA ferrimagnetic alloys. Alternatively, the authors could explain why HHVR (despite the ongoing controversy) is still more reliable than the hysteresis loop shift method.*

Response: Following the suggestion of the referee, we have added on page 3 of our manuscript that “We choose the HHVR technique with excitation of a *sinusoidal* electric field E (typically 30 kV/m) because it allows very accurate, consistent determination of spin-orbit torques for both IMA and PMA samples³⁰⁻³² without introducing significant thermal heating (Fig. 1a,b and Supplementary Fig. S4). ... $j_c = E/\rho_{xx}$ is the *sinusoidal* current density in the Pt_{0.75}Ti_{0.25} with resistivity ρ_{xx} ($j_c \approx 2.2 \times 10^6$ A/cm² for $E = 30$ kV/m).” As we have shown in our manuscript, there is negligible anomalous Nernst effect during our HHVR measurements (Fig.S4 in the Supplementary Information) and Hall voltage hysteresis loops indicates coercivity (perpendicular depinning field) and squareness that are fairly close to that from SQUID measurements (Fig. 1a,b). As we had discussed in the manuscript, “The “planar Hall correction” is negligible for these PMA Fe_xTb_{1-x} samples ($V_{Ph}/V_{AH} < 0.1$, see Fig. S4 in in the Supplementary Information)”.

The **dc current**-driven loop shift technique is not a good option for studying nearly compensated ferrimagnets (but seems to work for some conductive 3d ferromagnet and ferrimagnets far from the compensation). This is because loop shift measurement requires *very large dc currents* constantly applied to the Hall bars to observe enough loop shift, which can lead to long-time, considerable thermal heating. The latter considerably changes the temperature, magnetization, and ferrimagnetic compensation configuration during the loop-shift measurements and affects the calculation of spin torque efficiency. DC heating can cause irreversible degradation of the magnetic properties of near-compensation ferrimagnetic sample that are usually not very stable to large thermal heating. Existing papers including Ref. 64 have typically indicated rather strong uncertainties for the SOT efficiencies from loop-shift measurements. On page 8 of our revised manuscript, we have added that “The loop-shift measurements in Ref. 64 also applied large *dc* current densities of $\sim 10^7$ A/cm², close to the switching current density, in the resistive Ta/Co_xTb_{1-x} samples, leading to considerable Joule heating that had significantly altered the temperature, M_s , and ferrimagnetic compensation points of those samples. The latter resulted in additional uncertainties in those loop-shift results of ξ_{DL}^j and ultimately prevented resolving the variation of ξ_{DL}^j with the bulk properties of FIMs (i.e., $\tau_M^{-1}/\tau_{so}^{-1}$) in that work”.

We had also tried spin-torque ferromagnetic resonance which we have access and find no resonance response from the amorphous FeTb samples due to their very large damping, resistivity, and anisotropy. One might have also considered current-induced magnetization switching technique, but that technique

cannot even give a qualitative guidance of the relative strengths of spin-orbit torques of different samples (see Phys. Rev. Appl. 15, 024059 (2021)).

Comment A-8: 5) Related to the point above, how reliable are the three HHVR-derived SOT efficiency data points for the in-plane anisotropy samples? Can they be compared on the same ground as the data for the PMA samples?

Response: We have added on page 3 of our manuscript that “We choose the HHVR technique with excitation of a sinusoidal electric field E (typically 30 kV/m) because it allows very accurate, consistent determination of spin-orbit torques for both IMA and PMA samples³⁰⁻³²...” The reasonable consistence of the in-plane and out-of-plane HHVR results have been discussed based three independent material systems in our previous papers Ref. 30-32 and summarized again in our recent work arXiv:2207.05968 (in press at Phys. Rev. Applied). See below for several examples on the consistence of in-plane HHVR results and out-of-plane HHVR results of damping-like spin-orbit torque efficiencies from Ref. 30-32:

Fig. R1 Consistency of in-plane HHVR results and out-of-plane HHVR results of dampinglike spin-orbit torque efficiencies per unit current density for (a) $\text{Pd}_{1-x}\text{Pt}_x/\text{Co}$ or $\text{Fe}_{0.6}\text{Co}_{0.2}\text{B}_{0.2}$ bilayer with different Pt concentration in the $\text{Pd}_{1-x}\text{Pt}_x$ [30], (b) $\text{Au}_{1-x}\text{Pt}_x/\text{Co}$ bilayer with different Pt concentration in the $\text{Au}_{1-x}\text{Pt}_x$ [31], and (c) $[\text{Pt } d/\text{Hf } 0.2]_n/\text{Pt } d/\text{Co}$ with different thickness of each Pt slide [32]. The “IMA” represents the in-plane harmonic Hall voltage response results from in-plane magnetized samples, while the “PMA” represents the out-of-plane harmonic Hall voltage response results from perpendicular magnetic anisotropy samples, with NO so-called “planar Hall correction”. Clearly, the out-of-plane HHVR results without “planar Hall correction” are reasonably close to that of in-plane HHVR results.

Comment A-9: 6) The authors’ claim implies that it is very difficult, or perhaps impossible, to switch metals with compensated antiferromagnetic order. This would be very bad news for the field of antiferromagnetic spintronics. I am not sure whether the authors want to convey the bad news more explicitly (than what they already say toward the bottom of Page 7), but I feel that this is another big far-reaching implication of this present study.

Response: This an encouraging suggestion and we have added Figure 4 and extended discussions in our revised manuscript to show the implications of our finding for antiferromagnets.

“We also note that in principle a spin current still has the potential to switch the local nonzero atomic magnetizations within a perfectly compensated FIMs, single-layer AFs, and synthetic AFs (e.g., FM/Ru/FM) by local SOT on each magnetic site (Fig. 4b-d). However, unfortunately such switching is very inefficient in overcoming the switching barriers of the samples (e.g., magnetic anisotropy, pinning field, damping, etc) since the spin-orbit scattering easily overwhelms over spin transfer to the magnetization in the relaxation

process of incident spin current. It is rather typical that the effective magnetic anisotropy fields and pinning fields (i.e., the coercivities in the magnetization hysteresis) of hard rare-earth transition-metal FIMs and synthetic AFs¹¹ turn to diverge at the magnetization compensation point, **making them unswitchable for a realistic spin current**. Collective switching of the Néel vector of AFs also requires net SOT to overcome the barriers against switching (e.g., magnetic anisotropy, pinning, damping, etc), which is challenging to achieve, particularly so in present of spin-orbit scattering. Therefore, fully compensated FIMs and AFs themselves are most likely not an option as “Néel-vector-type” free layer of magnetic memory in terms of electrical writability and readability.”

Comment A-10: 7) *Related to the point above, Refs 12 and 13 are on insulating antiferromagnets. If space permits, it may be worth pointing out that the mechanisms of transverse spin current transport and relaxation are likely different between metallic and insulating ferri/antiferromagnets.*

Response: Following the referee’s suggestion, we have added in our revised manuscript that

Page 5: “Note that the large and robust $G_{\text{eff}}^{\uparrow\downarrow}$ for electron-mediated spin transport at the metallic $\text{Fe}_x\text{Tb}_{1-x}$ interface is in sharp contrast to that of ferrimagnetic insulator interfaces^{46,47} where thermal magnons mediate the spin transport such that a very low magnetic moment density reduces $G_{\text{eff}}^{\uparrow\downarrow}$.”

Page 8: “This generalized equation should apply to FMs, FIMs, and AFs that are metals or insulators. In magnetic insulators, an incident spin current carried by magnons transfers angular momentum to the magnetization via exchange interaction (with spin relaxation rate τ_M^{-1}) and also to the lattice via spin-orbit scattering of spin carriers (with spin relaxation rate τ_{so}^{-1}).”

Reviewer #2 (Remarks to the Author):

Comment B-1: *This manuscript reports the spin-orbit-torque (SOT) measurement in ferrimagnetic TbFe. There are many reports on the SOT in ferrimagnets, but some discrepancies appear among the reports thus far. This is because the ferrimagnets, in particular the rare earth-transition metal (RE-TM) ferrimagnets, are very sensitive to the composition and temperature, and even to the oxidation. Therefore, it is highly desirable to get a comprehensive understanding on the SOT in ferrimagnets.*

This work presents systematic studies on the SOT in ferrimagnets, which can be accomplished by changing the composition and temperature in a controlled manner. Based on their observations, the authors propose a more complete description about the SOT in ferrimagnets. They claim that the SOT is proportional to the (exchange-induced angular momentum transfer rate/spin-orbit-scattering loss rate). More simply speaking, the SOT is proportional to the magnetization (M_s), and thus the SOT would be negligible in antiferromagnet. Furthermore, the SOT is inversely proportional to momentum scattering rate, which suggests that the high SOC material is disadvantageous for SOT switching.

This kind of result can certainly advance our understanding by providing a deep physical insight, and therefore will lead a community. At the same time, however, it could mislead the community if the conclusion is made based on weak evidences. After reading the manuscript, I am still not fully convinced that the authors claimed in this manuscript because of the following reasons.

Response: We first thank the referee for the very helpful comments.

Comment B-2: *1. There are many reports about the existence of SOT in antiferromagnetic systems (e.g., DuttaGupta et al., Nat. Commun. 11, 5715 (2020), Kim et al., PRB 104, 054406 (2021), Tsai et al., AIP advances 11, 045110 (2021), etc). Do these results compatible with the authors' claim?*

Response: Thanks for drawing our attention to the three references.

Nat. Commun. 11, 5715 (2020), which is Ref. 74 in our manuscript, indicated in its abstract that “spin currents arising in the collinear IrMn layer exert SOTs on uncompensated antiferromagnetic moments”. The net uncompensated moments behavior as a ferromagnet and can certainly switched by spin current. As we have discussed on page 9 of our revised manuscript: “**The previous observation of current-driven switching of uncompensated magnetic domains or magnetizations embedded within AF hosts using transport or imaging methods^{59-62,74} (which is not real switching of AF Néel vector) is naturally explained by the small but sizable SOTs in nearly but not fully compensated systems..**”

PRB 104, 054406 (2021) report numerical simulation of a canted antiferromagnet with biaxial easy anisotropy. That work assumed a weak magnetization \mathbf{m} (along y direction) that is induced by DMI and thus perpendicular to the Neel vector (along x direction). They suggest that a spin current polarized along y direction can exert damping-like torque on \mathbf{m} , which leading to *tilting* of the Neel vector that is coupled to magnetization via DMI. Such tilting effect is apparently not a 180 reversal of magnetic moment or Neel vector and thus is not directly relevant to the topic and physics of our present manuscript.

AIP Advances 11, 045110 (2021) reports rotation of “magnetic octupole” between adjacent two of the sixfold easy axes of *non-collinear* antiferromagnet Mn_3Sn by magnetic field and transverse spin current. As far as we understand, the claim for Mn_3Sn was that: first apply a large magnetic field to align the magnetic moment to one easy axis (at 30°), then turn on a spin current to balance the moment to 0° or 60° via torque, then reverse the spin polarization will switch the magnetic moment between 0° and 60° while the magnetic field is kept on at 30° (the loop they show is from this $0^\circ \leftrightarrow 60^\circ$ rotation); when the spin current is turned off while the magnetic field is fixed at 30° (-30°), the magnetic moment will relax to the easy axis

at 30° (-30°). please refer to Nature 580, 608(2020) for a better explanation. Such switching is apparently not a 180 reversal of magnetic moment or collinear Neel vector and thus is not related to the topic and physics of our present paper.

Comment B-3: 2. *In the experiment, the magnetization varies orders of magnitude (from 1500 emu/cc to nearly zero), but the spin-orbit scattering changes at most factor 10. It seems that the results can be explained solely by the variation of magnetization. Why do they think the bulk spin-orbit-scattering is a key to understand their results?*

Response: If the referee could more carefully read our discussion on page 3, he/she would find that the damping-like torque does not vary in proportion or in a monotonic manner with the net magnetization: in the composition series, “ **M_s decreases monotonically by a factor of 33... ξ_{DL}^j decreases by a factor of 7 at 300 K.**” In the temperature series, “ **M_s and ξ_{DL}^j for the Pt_{0.75}Ti_{0.25}/Fe_{0.59}Tb_{0.41} sample are tuned by > 2 times and by > 7.5 times, respectively.**”

In our revised manuscript, we have also added Fig. 2e,f and discussion that “**Apparently, ξ_{DL}^j correlates closely with net magnetization (Fig. 2a,b) but not in a proportion or monotonic manner (Fig. 2e,f), suggesting a rather critical role of the net magnetization and another bulk effect (which, as we discuss later, is spin-orbit scattering) in the determination of ξ_{DL}^j .**”

We have made the physics more obvious using the schematic in Fig. 4a: the damping-like torque of the metallic heterostructures will not change with magnetization if the bulk spin-orbit scattering is absent; the damping-like torque would vary with magnetization only if there is significant spin-orbit scattering. Thus, the strong variation of damping-like torque with composition and temperature can only be understood by taking into account both the competing exchange interaction and spin-orbit scattering effects.

We would like to note that ref. 47 observed a thickness dependence of SOTs in a ferrimagnetic insulator related to thickness dependent magnetization. ***In our view***, there were three mechanisms at work in that case: (1) the *thermal magnon*-mediated spin mixing conductance of the magnetic insulators may be sensitive to the very low density of magnetic moment at the interface (which is possible, as indicated by theory in ref. 46); (2) the thickness variation is important as it is not much greater than the spin relaxation length which is long due to the very low magnetization and damping; (3) the increase of magnetization with thickness leads to enhancement of $\tau_M^{-1}/\tau_{so}^{-1}$ and thus torque with thickness. The first two mechanisms are clearly not at work in the case for our metallic FeTb samples. As we have determined on page 4 of the manuscript, the electron-mediated spin-mixing conductance of our FeTb interface is robust to composition and temperature, and the thickness of 8 nm is greater than the spin relaxation length set by spin dephasing and spin-orbit scattering together.

Comment B-4: 3. *I feel that the analysis on the spin-orbit-scattering shown in Fig. 3 and sec. 6 of SI is a bit clumsy. More rigorous analysis is required by considering the followings.*

3-1. *They used simple Drude model to get the transport parameters. Do they confident that the Drude model can be applicable to the RE-TM ferrimagnets which are amorphous and highly resistive compared to the normal metals?*

3-2. *They assumed that the electron effective mass was equal to the free electron mass. However, the electron effective mass in solid is different to the free electron mass. Furthermore, the effective mass in solid is strongly dependent on the band structure as well as on the*

temperature. As they changed the composition and temperature of TbFe, the effective mass should have some variation.

Response: Stimulated by the referee's comment, we have added in the Supplementary Note 5 that "Previous experiments⁷⁹ and theories⁵⁶ have suggested that the conduction of the disordered Fe_xTb_{1-x} is dominated by holes of the *d* bands and that the Drude model provides a realistic description for the relation of resistivity and *n* of the disordered Fe_xTb_{1-x}."

We have also noted on page 7 of our revised manuscript that "the value of τ_e is a rough estimate from the resistivity of the Fe_xTb_{1-x} following the Drude model" and that "We note that the applicability of the Drude model, the approximated value of m^* ... for estimating τ_e^{-1} is not essential for our conclusion of the strong variation of ξ_{DL}^j with relative spin relaxation rates, since similar scaling in Fig. 3e is present even when simply plotting ξ_{DL}^j as a function of $M_s/\zeta_{so}\rho_{xx}$ (Supplementary Fig. S11)."

At the beginning of Supplementary Note 5, we had noted that "We make a rough estimate of the momentum-scattering time (τ_e) of the Fe_xTb_{1-x} from the resistivity...."

3-3. More detailed description about how they subtracted the ordinary Hall effect of PtTi is required.

Response: Following the referee's suggestion, we have added in Supplementary Note 5 of our revised Supplementary Fig. S9c and discussions that

"We then measured the $dV_{1\omega}/dH_z$ values for the Pt_{0.75}Ti_{0.25} 5.6 nm/Fe_xTb_{1-x} 8 nm bilayers and for a 5.6 nm Pt_{0.75}Ti_{0.25} control sample from the slope of the linear fit of $V_{1\omega}$ vs H_z (Fig. S9a,b) in the high field regime by applying the same electric field *E*. As shown in Fig. S10, $dV_{1\omega}/dH_z$ of the 5.6 nm Pt_{0.75}Ti_{0.25} shows little temperature dependence and is negligibly small compared to that of the Pt_{0.75}Ti_{0.25} 5.6 nm/Fe_xTb_{1-x} 8 nm bilayers, suggesting that the $dV_{1\omega}/dH_z$ values of the Pt_{0.75}Ti_{0.25} 5.6 nm/Fe_xTb_{1-x} 8 nm bilayers are dominantly contributed by the ferrimagnetic Fe_xTb_{1-x} layer at all temperatures. After subtracted the small $dV_{1\omega}/dH_z$ contribution from that of the Pt_{0.75}Ti_{0.25} 5.6 nm/Fe_xTb_{1-x} 8 nm bilayers, we obtain the $dV_{1\omega}/dH_z$ values for the Fe_xTb_{1-x} layers. The ordinary Hall coefficient (R_{OH}) of the Fe_xTb_{1-x} is then determined following the relation $R_{OH} = (\rho_{xx}/EW) dV_{1\omega}/dH_z$, where ρ_{xx} is the resistivity and *W* the width of the Hall bar."

3-4. In Fig. S10(b), the ordinary Hall resistance seems to rapidly increase at higher temperature. How do they explain this? Is it possible to come from the paramagnetic effect with approaching Curie temperature?

Response: The temperature dependence of ordinary Hall coefficient (R_o) is most likely due to a temperature tuning of Fermi surface properties and cannot be explained by any paramagnetic response. Similar increase of R_o with increasing temperature has been reported for disordered Fe (that has a very high Curie temperature) and Tb crystals below their magnetic ordering temperature (pure Tb is a low-Neel-temperature antiferromagnet) due to variation of Fermi surface properties (see the first figure below as reprinted from Ref. 54 and Ref.55).

In contrast, there is no indication of paramagnetic response below the Curie temperature of our FeTb samples ($T_c \approx 400$ K for the sample in Fig. S9d). At high magnetic fields, the Hall voltages of the $\text{Pt}_{0.75}\text{Ti}_{0.25}/\text{Fe}_x\text{Tb}_{1-x}$ samples are good linear function of magnetic field in the whole studied temperature range, suggesting a magnetic-field-independent ordinary Hall coefficients (see Supplementary Fig. S9b). Moreover, R_o increases in the whole temperature region and cannot be explained by any effects near the Curie temperature.

In Supplementary Note 6 we have added Fig. S9b,c and discussion that “As shown in Fig. S9d, R_{OH} of the $\text{Fe}_x\text{Tb}_{1-x}$ increases strongly with raising temperature, which is consistent with previous reports of strong variation of R_{OH} with temperature in Fe and Tb due to temperature tuning of Fermi surface properties. Note that at high magnetic fields, the Hall voltages of the $\text{Pt}_{0.75}\text{Ti}_{0.25}/\text{Fe}_x\text{Tb}_{1-x}$ samples are fairly good linear function of magnetic field in the whole studied temperature range, suggesting a magnetic-field-independent ordinary Hall coefficients (see Fig. S9b for the 350 K data for the $\text{Pt}_{0.75}\text{Ti}_{0.25}/\text{Fe}_{0.59}\text{Tb}_{0.41}$). The increase of R_o in the whole temperature region cannot be explained by any effects due to approaching the Curie temperature (400 K for the $\text{Pt}_{0.75}\text{Ti}_{0.25}/\text{Fe}_{0.59}\text{Tb}_{0.41}$). ... As suggested by the significant variations in the anomalous Hall voltage of $\text{Fe}_x\text{Tb}_{1-x}$ as a function of composition and temperature (Fig. S10), the Fermi surface properties of the $\text{Fe}_x\text{Tb}_{1-x}$ are strongly tuned by the composition and temperature. The competition of the Tb 5d and 6s electrons and the Fe 3d and 4s electrons at the Fermi surface appears to explain the moderate and non-monotonic change of the carrier density as a function of the composition of $\text{Fe}_x\text{Tb}_{1-x}$. Rigid understanding of the variations of the carrier density requires precise calculation of the band structure for the amorphous $\text{Fe}_x\text{Tb}_{1-x}$ and is beyond the scope of our present work.”

3-5. *It is generally understood that the carrier density strongly depends on the band structure, which means that the carrier density is more sensitive to the composition variation rather than the temperature change. However, Fig. S10(d) shows that the amount of the variation of carrier density is almost same for both cases.*

Response: Stimulated by the referee’s question, we have added in our revised Supplementary Information that “As shown in Fig. S9d, R_{OH} of the $\text{Fe}_x\text{Tb}_{1-x}$ increases strongly with raising temperature, which is consistent with previous reports of strong variation of R_{OH} with temperature in Fe and Tb due to temperature tuning of Fermi surface properties... As suggested by the significant variations in the anomalous Hall voltage of $\text{Fe}_x\text{Tb}_{1-x}$ as a function of composition and temperature (Fig. S10), the Fermi surface properties of the $\text{Fe}_x\text{Tb}_{1-x}$ are strongly tuned by the composition and temperature. The competition of the Tb 5d and 6s electrons and the Fe 3d and 4s electrons at the Fermi surface appears to explain the moderate and non-monotonic change of the carrier density as a function of the composition of $\text{Fe}_x\text{Tb}_{1-x}$.”

Importantly, we have added on Page 7 of our revised manuscript that “We note that the applicability of ... the single-band model for the ordinary Hall effect for estimating τ_e^{-1} is not essential for our conclusion of the strong variation of ξ_{DL}^j with relative spin relaxation rates, similar scaling in Fig. 3e is present even when simply plotting ξ_{DL}^j as a function of $M_s/\zeta_{so}\rho_{xx}$ (Supplementary Fig. S12).

Comment B-5: TbFe is known as a “sperimagnet”, in which the Tb and Fe moments are spread out and form a cone angle due to the strong anisotropy of Tb and weak exchange of Fe (Hebler et al., Fron. Mater. 3, 8 (2016) doi: 10.3389/fmats.2016.00008). Recent report showed that the sperimagnetic properties caused exotic phenomena (Park et al., Nat. Commun. 13, 5530 (2022)). As the sperimagnet has a dispersed spin structure, it can also affect to the spin scattering. Can the authors exclude this possibility?

Response: As discussed in the reference suggested by the referee [Fron. Mater. 3, 8 (2016)], the so-called “sperimagnetism” means disorder-induced local distribution of atomic magnetizations, random in the film plane but preferable in the film normal for FeTb (see the Figure below). In the limit that the distribution of atomic magnetizations is completely random along all directions, the material will macroscopically have no net magnetization ($M_s = 0$) as a paramagnetic material for the spin current.

FIGURE 1 | (A) Magnetic moment distribution in an amorphous Tb-Fe material with two preferred antiparallely oriented magnetic sublattices. **(B)** Averaged distribution of the magnetic moments (sperimagnetism).

Fundamentally, the direct interaction of spin current with any individual or collection of atomic magnetizations is exchange interaction. Transfer of spin angular momentum from spin current exerts a net spin torque on the magnetic individual or collection and excite fast dynamics which will relax due to other subsequent mechanisms (e.g., spin-orbit scattering of angular momentum into lattice), while the relaxation of magnetization dynamics will not affect the quasi-static measurement of spin-orbit torque. Thus, we consider that local distribution of magnetizations is unlikely to provide any fundamental spin relaxation mechanism additional to spin-magnetization exchange interaction and spin-orbit scattering.

Stimulated by the referee’s comment, we have commented on page 7 in our revised manuscript that “In above discussions, we ignored any effect of local distribution of magnetizations (also known as sperimagnetism⁵⁸) because it, if present, may only indirectly affect the average spin relaxation rates of spin-magnetization exchange interaction and spin-orbit scattering via reducing the net magnetization and strengthening SOC-related momentum scattering of spin carriers, respectively.”

Comment B-6: 5. *The magnetic moment of Fe arises from the 3d orbital electron, while that of Tb is mainly determined by the 4-f orbital electron which lies far lower than the Fermi level. In this manuscript, the authors considered the Fe and Tb moments equally when considering the spin scattering. What is microscopic background in which conduction electrons interact with 4-f electrons through the exchange interaction and SOC?*

Response: First, atomic magnetizations of the Fe and the Tb are strongly exchange-coupled due to the ferrimagnetism in a quasi-static process such that the ferrimagnetic system act as a net magnetization to a travelling spin current when considering spin relaxation via the exchange interaction. This is similar to the interplay of a spin current with a ferromagnetic layer that is thicker than the spin relaxation length but well exchange-coupled. The spin current acts to torque and switch the whole magnetic layer, not only the first few layers near the interface, despite that micromagnetic dynamics excited by the spin-orbit torque might not be coherent for different spots in a very-short timescale (e.g. picosecond timescale).

In spin-orbit scattering process, spin current relaxes due to SOC-related momentum scattering of the travelling spin carriers (conduction electrons carrying spin polarization information) at the Fermi surface of the $\text{Fe}_x\text{Tb}_{1-x}$ alloy by impurity and others without involving any spin-magnetization exchange interaction. Also, both Fe and Tb contributes *s* and *d* electrons to the Fermi surface.

Comment B-7: <Minor comments>

1. p. 3, below the Fig. 2: “thermoelectric effects (see details in Sec. 1 in the Supplementary Information).” Sec. 1 should be Sec. 3

Response: We have corrected this typo, thanks for pointing that out.

Comment B-8: 2. *I recommend to denote the current density that they used in their experiments.*

Response: Thanks for the suggestion. During the harmonic Hall voltage response measurement in this work, we always apply an electric field E to the Hall-bar devices as the excitation (e.g., 30 kV/cm), such that the current density is different for different layers and for different $\text{Fe}_x\text{Tb}_{1-x}$ samples. More importantly, use of current density can be misleading because for most of the measurements in this work (e.g. Fig. S2-S4) both current density in the PtTi and FeTb are involved and important. In some other case, only the current density in the magnetic layer is meaningful for Fig. 1b,d. Therefore, we only note that electric field in our measurement and also the resistivities of each layer. The current density in each layer in each measurement is simple to know from the Ohm’s law, $j = E/\rho_{xx}$.

In our revised manuscript, we have noted that “**We calculate the SOT efficiency using $\xi_{\text{DL}}^j = (2e/\hbar)M_s t_{\text{FeTb}} H_{\text{DL}}/j_c$, where $\dots j_c = E/\rho_{xx}$ is the current density in the $\text{Pt}_{0.75}\text{Ti}_{0.25}$ with resistivity ρ_{xx} ($j_c \approx 2.2 \times 10^6 \text{ A/cm}^2$ for $E = 30 \text{ kV/m}$).**”

Comment B-9: 3. *In Fig. S5(a), mark the origin (zero field point) on the graph.*

Response: We have marked the origin in Fig. S4a (originally Fig. S5a), thanks for pointing that out.

Comment B-10: 4. *In Fig. S5(b)-(f), why does the 2nd harmonic signal have some offset for zero field?*

Response: A main source of the zero-field offset is magnetic multidomain state. When an in-plane magnetic field is swept from larger negative to large positive, perpendicular magnetic anisotropy (PMA) samples may fall into multidomain state that contributes to additional second harmonic signals at the low fields. In Fig. S4b-f (it was Fig. S5b-f) we only need the very high field data to determine the anomalous Nernst voltage due to the *perpendicular* thermal gradient ($V_{ANE,z}$, different from $V_{ANE,x}$). Of course, we never use such large field range scans like those in Fig S5 to determine the strength of SOTs. In Supplementary Note2 of the Supporting Information, we have clearly noted that “During the measurements, we first apply a large out-of-plane magnetic field to saturate the sample and to avoid any intermediate or multi-domain states, and then record HHVR while sweeping $H_{x(y)}$ only in the small-field region (≤ 3.5 kOe).” and that “Note that the data in Fig. S4 were collected by sweeping the magnetic field along the in-plane current direction between ± 9 T. The slopes in Fig. S4 at low fields might therefore be affected by non-uniform, multi-domain magnetic states and should not be used to estimate the values of $H_{DL(FL)}$.”

Another possible source of the zero-field offset of the second HHVR in such high field scans is the anomalous Nernst voltage due to the *longitudinal* thermal gradient ($V_{ANE,x}$) (please refer to Eq. S6 of the Supplementary Information of Appl. Phys. Rev. 9, 021402 (2022)). However, $V_{ANE,x}$ is independent of in-plane magnetic field and will not affect the determination of SOT fields since $H_{DL(FL)} = -2 \frac{\partial V_{2\omega}}{\partial H_{x(y)}} / \frac{\partial^2 V_{1\omega}}{\partial H_{x(y)}^2}$.

Comment B-11: 5. Fig. S7(a) is inconsistent with other data. According to the authors’ claim, the damping-like effective field should be proportional to the magnetization. However, it seems that the data in Fig. S7(a) is somewhat opposite.

Response: The referee seems to have misunderstood our discussion. We have never claimed or suggested that the damping-like spin-orbit torque effective field (H_{DL}) were proportional to the magnetization. If that was true, then the torque efficiency ξ_{DL}^j would scale with M_s^2 , which is clearly not what we indicate in our work. Instead, we had clearly discussed in our Supplementary Information that “As shown in Figs. S7(a) and S7(b), H_{DL} of the $Pt_{0.75}Ti_{0.25}$ (5.6 nm)/ Fe_xTb_{1-x} (8 nm) does not scale linearly with $1/M_s$ or diverge upon approaching the magnetization compensation points in the composition series and the temperature series. This reveals that the damping-like SOT efficiency (ξ_{DL}^j) is not constant but rather varies with composition and temperature.”

Reviewers' Comments:

Reviewer #1:

Remarks to the Author:

I thank the authors for carefully addressing each comment. The authors have satisfactorily resolved my concerns and questions. In my view, they have also addressed satisfactorily the other referee's comments. I believe the paper can proceed toward publication.

Optional considerations - It is nice that the authors now discuss the implications for antiferromagnet switching in the revised paper. While reading it, the following thoughts came to my mind:

1) For "real" antiferromagnets, if I understand correctly, most device applications seek to switch the Neel vector by 90 degrees (distinguishable e.g. via AMR), rather than 180 degrees. Therefore, could it be argued that not all is lost yet for the prospect of antiferromagnetic spintronic memory devices?

2) If the spin dephasing length scale is extended in an antiferromagnet (and also scattering is suppressed), would perhaps a sizable field-like spin torque arise (while the damping-like torque is weak)? I wonder if the field-like torque may still be useful for 90-degree switching. But then, perhaps it's not great because it would have to overcome a sufficiently large anisotropy energy barrier?

Reviewer #2:

Remarks to the Author:

I agree with the authors that the spin-orbit scattering process should be taken into account for the complete understanding of SOT. However, I am still not convinced whether the present experimental data can support their claim.

The main conclusion of this study is that the SOT efficiency depends on the relative ratio between the relaxation rate into magnetization (τ_M^{-1}) and relaxation rate into lattice (τ_{so}^{-1}). They argue that the τ_M^{-1} is proportional to M_s , and the τ_{so}^{-1} is proportional to the spin orbit coupling strength and momentum scattering rate.

They mentioned that "it is reasonable to expect τ_M^{-1} is proportional to M_s for such ferrimagnetic alloys considering the cancelling effects of the exchange fields from the antiferromagnetically-aligned magnetic sub-lattices.", which I agree. However, the same magnetization can be obtained as the temperature approaches to Curie temperature as well as to compensation temperature (for example, $T=325K$ and $T=225K$ in Fig. 2f). Although the magnetization is same for both temperatures, their microscopic details are different. Can they argue that the relaxation rate only depends on the magnetization irrespective of the microscopic details?

The discussion of Fig. 3 is still unsatisfactory. I do not think such a rough estimation without experimental verification can support any scientific argument. The extraction of physical parameters has been done based on rough estimation, as the authors mentioned in their revised manuscript. Therefore, the concerning about the validity of such estimations still remains.

- The resistance of sample increases with reducing temperature (Fig. S9(e)), which is typical behavior for amorphous materials where the electron hopping appears. Therefore, I am still concerning whether the Drude model can be applicable in this sample (the reference 56 and 79 is not accessible to me).
- The variation of scattering time is much larger in temperature series compared to that in the composition series. In general, the band distortion can be more enhanced when we change the composition rather the temperature. Can they explain why the variation of scattering time for temperature series is larger than that for the composition series?
- The scattering time increases with increasing temperature (Fig. 3b), which the authors ascribe to

the electron-electron interaction or magnetic Brillouin zone scattering. More detailed discussion would be helpful since ref. 56 is not accessible to me (also to other potential readers).

- I think that the linear variation of spin-orbit scattering strength as a function of Fe composition (Fig. 3c) is also a "too strong" assumption. Other supporting calculation or relating literatures are required.

- Based on above considerations, I don't think the Fig. 3 can support the authors' argument.

They mentioned that the estimation is not essential for their conclusion by providing 'less-assumed' figure (Fig. S11). However, I see that almost of the data in Fig. S11 is gathered around the y-axis. Can they provide any functional formula based on which one can determine the validity of model?

Response letter to the 2nd reports of the reviewers

Reviewer #1 (Remarks to the Author):

Comment A-1: *I thank the authors for carefully addressing each comment. The authors have satisfactorily resolved my concerns and questions. In my view, they have also addressed satisfactorily the other referee's comments. I believe the paper can proceed toward publication.*

Optional considerations - It is nice that the authors now discuss the implications for antiferromagnet switching in the revised paper. While reading it, the following thoughts came to my mind:

1) For "real" antiferromagnets, if I understand correctly, most device applications seek to switch the Neel vector by 90 degrees (distinguishable e.g. via AMR), rather than 180 degrees. Therefore, could it be argued that not all is lost yet for the prospect of antiferromagnetic spintronic memory devices?

2) If the spin dephasing length scale is extended in an antiferromagnet (and also scattering is suppressed), would perhaps a sizable field-like spin torque arise (while the damping-like torque is weak)? I wonder if the field-like torque may still be useful for 90-degree switching. But then, perhaps it's not great because it would have to overcome a sufficiently large anisotropy energy barrier?

Response: We are grateful to the reviewer for recommendation.

We also thank the reviewer for the "optional considerations" about the possibility of 90-degree rotation of Neel vector (so-called spin-flop transition) by field-like torque. Field-like spin-orbit torque field is very weak, typically of the order of 10^{-5} - 10^{-3} Tesla. In contrast, the spin-flop field for such 90 degree rotation of the Neel vector is typically tens of Tesla (e.g., 9 Tesla for antiferromagnetic MnF₂ and 42 Tesla for FeF₂, PRL 122, 217204 (2019)). Therefore, field-like spin-orbit torque is not useful for spin-flop transition.

On page 9 of our revised manuscript, we have added that "**We also note that field-like effective SOT field is too weak to reach the spin-flop field required for 90° rotation of Néel vector (typically of the order of 1-10³ kOe).**"⁷⁷

Reviewer #2 (Remarks to the Author):

Comment B-1: *I agree with the authors that the spin-orbit scattering process should be taken into account for the complete understanding of SOT. However, I am still not convinced whether the present experimental data can support their claim. The main conclusion of this study is that the SOT efficiency depends on the relative ratio between the relaxation rate into magnetization (τ_M^{-1}) and relaxation rate into lattice (τ_{so}^{-1}). They argue that the τ_M^{-1} is proportional to M_s , and the τ_{so}^{-1} is proportional to the spin orbit coupling strength and momentum scattering rate.*

Response: The reviewer commented in his/her first report that our main finding that spin-orbit scattering must be taken into account for the complete understanding of SOT “*can certainly advance our understanding by providing a deep physical insight, and therefore will lead a community.*” The reviewer has now agreed in this report that “*spin-orbit scattering process should be taken into account for the complete understanding of SOT.*” This means that the reviewer has accepted our main finding, which we appreciate very much. The *remaining questions* of the reviewer are about a few approximations in the first paragraph of page 7 that we introduce to provide as an additional, more quantitative discussions.

With this in mind, below we try our best to address all the reviewers’ *remaining questions*.

Comment B-2: *They mentioned that “it is reasonable to expect τ_M^{-1} is proportional to M_s for such ferrimagnetic alloys considering the cancelling effects of the exchange fields from the antiferromagnetically-aligned magnetic sub-lattices.”, which I agree. However, the same magnetization can be obtained as the temperature approaches to Curie temperature as well as to compensation temperature (for example, $T=325K$ and $T=225K$ in Fig. 2f). Although the magnetization is same for both temperatures, their microscopic details are different. Can they argue that the relaxation rate only depends on the magnetization irrespective of the microscopic details?*

Response: Yes, we expect that the **spin-magnetization** relaxation rate of the same sample should be the same at 325 K and 225 K because of the same magnetization. At least, the data of temperature series fits well to Eq. (3) in Fig. 3e. In our view, whether the magnetization is lowered due to composition or temperature, its ability of absorbing spin angular moment is lowered. Certainly, we would appreciate it if the reviewer could help to specify any difference for the spin-magnetization interaction.

Comment B-3: *The discussion of Fig. 3 is still unsatisfactory. I do not think such a rough estimation without experimental verification can support any scientific argument. The extraction of physical parameters has been done based on rough estimation, as the authors mentioned in their revised manuscript. Therefore, the concerning about the validity of such estimations still remains.*

B-3.1- *The resistance of sample increases with reducing temperature (Fig. S9(e)), which is typical behavior for amorphous materials where the electron hopping appears. Therefore, I am still concerning whether the Drude model can be applicable in this sample (the reference 56 and 79 is not accessible to me).*

Response: Figure S14b in our revised supplementary information has clearly excluded electron hopping because the variation of resistivity with temperature *deviates remarkably from the scaling of hopping*

conduction. In fact, it is known from condensed matter physics that electron hopping occurs in disordered insulators or oxide-metal granular films with extremely high resistivities (e.g., 10^6 - $10^9 \mu\Omega^{-1} \text{ m}^{-1}$ for quasicrystal AlPdRe in PRL 81, 4204 (1998), amorphous GeTe, and GeSb_2Te_4 annealed at 150 °C in Nat. Mater. 10, 202 (2011), 10^{11} - $10^{17} \mu\Omega \text{ cm}$ for the Pt-SiO₂ granular film with Pt concentration of 0.11 in Physica B 279, 341 (2000) and Adv. Phys. 24, 407 (1975)). Therefore, hopping conduction should NOT be expected in the metallic $\text{Fe}_x\text{Tb}_{1-x}$ with resistivity of only **31-265 $\mu\Omega \text{ cm}$** .

We cannot accept the reviewers' assertion that the Drude model is “*such a rough estimation without experimental verification*”. As we noted in Supplementary Note 5, “**Previous experiments⁸¹ and theories⁵⁷ have suggested that ... the Drude model provides a realistic description for the relation of resistivity and n of the disordered $\text{Fe}_x\text{Tb}_{1-x}$.**” Below are the discussions captured from refs. 57 and 82 (refs. 56 and 79 in the previous version):

Ref. 57: Meaden, Conduction Electron Scattering and the Resistance of the Magnetic Elements, Contemp. Phys. 12, 313-337 (1971)

A number of approximations are necessary, of which the first is the assumed suitability of the **nearly-free electron model** to describe the $E(k)$ relation for the conduction electrons. **It works well enough for this particular purpose because the outer shell electrons (i.e. 5d and 6s) overlap considerably in the rare-earths. One can therefore use a straightforward approach, based essentially on a free electron model**

Ref. 82: Connell and Bloomberg, “Amorphous Rare-Earth Transition-Metal Alloys”, Physics of Disordered Materials, edited by David Adler, Hellmut Fritzsche, Stanford R. Ovshinsky, Plenum Press 1985, pp 739-752.

In Fig. 2, we have also shown the temperature dependence of the resistivity of $\text{Tb}_{0.2}\text{Fe}_{0.8}$. In agreement with the Mooij correlation,⁵ the temperature coefficient of the resistivity is negative (i.e. $d \ln \rho / dT = -4 \times 10^{-4} \text{ K}^{-1}$) and $\rho > 150 \mu\Omega - \text{cm}$. While our results do not provide a clearer explanation of this interesting correlation, the knowledge of the density of states at the Fermi level allows us to make some estimates of the proximity of the minimum metallic conductivity in these alloys and to rule out one proposed conduction mechanism. **We can also show that the dc and optical conductivities are apparently closely described by a Drude model with a very short relaxation time.** Furthermore, the extraordinary Hall effect and the

B-3.2- *The variation of scattering time is much larger in temperature series compared to that in the composition series. In general, the band distortion can be more enhanced when we change the composition rather the temperature. Can they explain why the variation of scattering time for temperature series is larger than that for the composition series?*

Response: As we have discussed in our manuscript, “ τ_e of the $\text{Fe}_x\text{Tb}_{1-x}$ varies by a factor of ≈ 6 by composition and a factor of ≈ 7 by temperature, suggesting a significant tuning of the Fermi surface properties. **The increase of τ_e in the $\text{Fe}_x\text{Tb}_{1-x}$ metal with raising temperature likely originates from electron-electron interaction⁵⁶ ($\rho_{\text{FeTb}} \propto T^{1/2}$, see Fig. S14a) or magnetic Brillouin zone scattering (the periodic potentials due to antiferromagnetic alignment of the magnetic sublattices can produce an additional magnetic Brillouin zone, of smaller volume in k -space than the ordinary lattice potential, whose planes further incise and contort the Fermi surface⁵⁷).**”

Honestly, the authors do not find it surprising at all that both the temperature (at the composition of $\text{Fe}_{0.59}\text{Tb}_{0.61}$) and composition (at 300 K) affect the scattering time significantly in such ferrimagnets whose

Fermi surface properties (e.g., anomalous Hall conductivity and carrier density) are strongly dependent on both temperature and composition. From our literature investigation, a very strong tuning of the momentum scattering time by temperature is very common (e.g., 4-5 times for PbTe in the left figure below from ACS Appl. Energy Mater. 5, 7260 (2022); 20-70 times for Li-doped MnTe in the middle figure below from Sci. Adv. 5, aat9461 (2019); scattering time increased by 1.5 times within very narrow temperature of 0-30 K, which is equal to 15 times for 0-300 K temperature range, for LaNdSrCuO in the right figure below from Nature 595, 667 (2021)).

Because the reviewer did not specify in which magnetic metal and why momentum scattering time is tuned much less with temperature than with composition, we cannot comment on why that specific situation was different from or implied anything against the case for the FeTb samples in our present study.

B-3.3- *The scattering time increases with increasing temperature (Fig. 3b), which the authors ascribe to the electron-electron interaction or magnetic Brillouin zone scattering. More detailed discussion would be helpful since ref. 56 is not accessible to me (also to other potential readers).*

Response: Following the reviewer’s suggestion, we have added in the supplementary information the Supplementary Note 8 and Fig. S14a and verify that the temperature profile of the resistivity agrees with electron-electron interaction. Magnetic Brillouin zone scattering may also lead to resistivity upturn upon cooling but there is no formula developed for this mechanism such that we cannot test it by fitting the data.

In 1st paragraph, page 7 of the manuscript, we have also extended our discussion: “**The increase of τ_c in the $\text{Fe}_x\text{Tb}_{1-x}$ metal with raising temperature likely originates from electron-electron interaction⁵⁶ ($\rho_{\text{FeTb}} \propto T^{1/2}$, see Supplementary Fig. S14a) or magnetic Brillouin zone scattering (the periodic potentials due to antiferromagnetic alignment of the magnetic sublattices can produce an additional magnetic Brillouin zone, of smaller volume in k -space than the ordinary lattice potential, whose planes further incise and contort the Fermi surface⁵⁷). Both electron-electron interaction and magnetic Brillouin zone lead to additional electron scattering manifesting as a resistivity upturn upon cooling.^{56,57}**”

For the convenient of the editor and the reviewer, we have also attached Ref. 57 (i.e., Ref. 56 in our previous version of manuscript) with the related sentences being yellow as additional information for review.

B-3.4- I think that the linear variation of spin-orbit scattering strength as a function of Fe composition (Fig. 3c) is also a “too strong” assumption. Other supporting calculation or relating literatures are required.

Response: In our revised manuscript, we have added supporting reference for the linear dependence on the alloy composition of the spin-orbit coupling strength:

“The average SOC strength of the $\text{Fe}_x\text{Tb}_{1-x}$ is estimated as $\zeta_{\text{so}} \approx x\zeta_{\text{so,Fe}} + (1-x)\zeta_{\text{so,Tb}}$ following the linear dependence on alloy composition of the bulk SOC⁵⁸ and the theoretical values⁵⁹ of $\zeta_{\text{so,Fe}} = 0.069$ eV for Fe and $\zeta_{\text{so,Tb}} = 0.283$ eV for Tb...”

Note that Taylor expansion suggests that the first-order approximation for a smoothly and monotonically varying quantity is a linear function.

B-3.5- Based on above considerations, I don't think the Fig. 3 can support the authors' argument.

Response: Based on our responses above, we believe we have adequately addressed all the remaining questions of the reviewer. In our view, we hope the reviewer and the editors also agree now, the Drude model for the calculation of momentum scattering rate in Fig. 3b is reasonable approximation and well supported by references. It is already the best one that is doable for the authors and most likely the whole community. The analysis utilizing the Drude model describes the experimental data very well (Fig. 3e).

Finally, the very helpful Fig. 3b-e and the associated first paragraph of page 7 are NOT the *base* of our manuscript but rather *additional*. Even without Fig. 3b-e and the associated first paragraph of page 7, our main finding of the strong variation of the spin-orbit torque with the relative spin relaxation rates and all the other discussions hold well. The variation of the spin-orbit torque is readily measured in Fig. 2 of our manuscript, and we have excluded all the other mechanisms on pages 4-6 of our manuscript. Therefore, there is no point to tangle about the calculation of the momentum scattering time anymore.

B-3.6- They mentioned that the estimation is not essential for their conclusion by providing 'less-assumed' figure (Fig. S11). However, I see that almost of the data in Fig. S11 is gathered around the y-axis. Can they provide any functional formula based on which one can determine the validity of model?

Response: We simply display the experimental truth in Fig. S11. The strong variation of the spin-orbit torque and the ratio of $\zeta_{\text{so}}\rho_{xx}$ (correlated to spin-orbit scattering) and $1/M_s$ (correlated to the relaxation via spin-magnetization exchange interaction) is *readily* clear from the experimental data in Fig. S11. This is simply the truth and does not need any formula to fit or to validate any model. Does it?

Ps: The data points in Fig. S11 are more concentrated near the low $M_s/\zeta_{\text{so}}\rho_{xx}$ region because variation of carrier density with the composition and temperature is not subtracted yet from $M_s/\zeta_{\text{so}}\rho_{xx}$ (but has been subtracted from $M_s/\zeta_{\text{so}}\tau_e^{-1}$ in Fig. 3e). Even so, strong variation of the spin-orbit torque with the relative scattering rate is so robust that it is readily clear in Fig. S11. The reviewer must agree that it is the consensus for the spintronics community that the spin-orbit torque would be a constant in the similar case if the relative spin relaxation rates played no role (the red line in Fig. 4a of our manuscript).